# Trapping and detecting nanoplastics by MXene-derived oxide microrobots

Mario Urso [1], Martina Ussia [1], Filip Novotný[1,2] & Martin Pumera [1,2,3,4 ✉]

Nanoplastic pollution, the final product of plastic waste fragmentation in the environment, represents an increasing concern for the scientific community due to the easier diffusion and higher hazard associated with their small sizes. Therefore, there is a pressing demand for effective strategies to quantify and remove nanoplastics in wastewater. This work presents the "on-the-fly" capture of nanoplastics in the three-dimensional (3D) space by multifunctional MXene-derived oxide microrobots and their further detection. A thermal annealing process is used to convert $Ti_3C_2T_x$ MXene into photocatalytic multi-layered $TiO_2$, followed by the deposition of a Pt layer and the decoration with magnetic $\gamma$-$Fe_2O_3$ nanoparticles. The MXene-derived $\gamma$-$Fe_2O_3$/Pt/$TiO_2$ microrobots show negative photogravitaxis, resulting in a powerful fuel-free motion with six degrees of freedom under light irradiation. Owing to the unique combination of self-propulsion and programmable Zeta potential, the microrobots can quickly attract and trap nanoplastics on their surface, including the slits between multi-layer stacks, allowing their magnetic collection. Utilized as self-motile preconcentration platforms, they enable nanoplastics' electrochemical detection using low-cost and portable electrodes. This proof-of-concept study paves the way toward the "on-site" screening of nanoplastics in water and its successive remediation.

[1] Future Energy and Innovation Laboratory, Central European Institute of Technology, Brno University of Technology, Purkyňova 123, 61200 Brno, Czech Republic. [2] Center for Advanced Functional Nanorobots, Department of Inorganic Chemistry, Faculty of Chemical Technology, University of Chemistry and Technology Prague, Technická 5, 166 28 Prague, Czech Republic. [3] Department of Chemical and Biomolecular Engineering, Yonsei University, 50 Yonsei-ro, Seodaemun-gu, Seoul 03722, Korea. [4] Department of Medical Research, China Medical University Hospital, China Medical University, No. 91 Hsueh-Shih Road, Taichung, Taiwan. ✉email: martin.pumera@ceitec.vutbr.cz

The images of marine environments full of plastic bags, bottles, and other plastic waste are impressed in our minds and mirror humankind's unpreparedness to manage them[1,2]. Unfortunately, the actual hazard of plastics is not restricted only to what is visible to our eyes. Plastic materials fragment into smaller pieces with sizes below 5 mm, called microplastics[3]. These can further break down into even smaller and more dangerous pieces (1–1000 nm), referred to as nanoplastics[4–6]. In fact, microplastics typically sediment on the seafloor, while nanoplastics remain suspended in water due to their lower weight[7]. Then, they are transported by the ocean currents, diffusing in short times. Because of their high surface-to-volume ratio, nanoplastics can absorb large amounts of toxic pollutants in water and serve as the substrate for the growth of pathogenic bacterial biofilms, increasing their toxicity[7,8]. Contrarily to the microplastics, they can easily penetrate tissues, posing serious risks to the health of all living beings[9].

The detection of nanoplastics in water samples and their consequent removal is critical. Scanning electron microscopy (SEM) and transmission electron microscopy (TEM) allow for visualizing nanoplastics but do not provide other information about the plastic material[10]. Similarly, nanoparticle tracking analysis (NTA) measures the nanoplastics' size distribution and concentration by recording the scattered light from an incident light beam[11]. Mass spectrometry techniques are also promising for studying nanoplastics. In this regard, Mitrano and coworkers synthesized nanoplastics with a metallic core to monitor their fate in the environment through inductively coupled plasma mass spectrometry (ICP-MS)[12]. However, a strategy for the rapid and "on-site" screening of nanoplastics in water samples without the need for expensive laboratory instruments and specialized staff is missing[13]. Besides, the remediation of nanoplastics-contaminated waters is crucial. Conventional approaches for removing microplastics, such as filtration, are not suitable for nanoplastics due to their tiny size[14]. On the other hand, the concept of microplastics' capture by electrostatic forces using oppositely charged magnetic particles and their successive collection with magnets can be potentially extended to the nanoplastics[15].

Micro/nanorobots, combining the unique physicochemical properties of micro/nanoscale materials with the autonomous motion ability and programmable functionalities, are revolutionizing all application fields, including environmental remediation[16–18], sensing[19,20], biomedicine[21,22], and electronics[23]. Specifically, in water remediation, their active movement overcomes the limitation of passive diffusion and enhances the interaction with pollutants, accelerating the purification process[24]. Among the different micro/nanorobots, light-powered ones are particularly promising for this purpose since light is an abundant and powerful energy source to induce their motion and, at the same time, favor the photocatalytic degradation of pollutants through advanced oxidation processes[25]. The most investigated light-powered micro/nanorobot consists of a Janus particle formed by a photocatalytic semiconductor ($TiO_2$, ZnO, $Fe_2O_3$) asymmetrically covered by a metal layer (Pt, Au)[26–28]. The metal layer deposition can be avoided for some intrinsically asymmetric microrobots ($BiVO_4$, $Bi_2WO_6$), but they require small amounts of toxic $H_2O_2$ fuel to move under light irradiation[29,30]. In the last years, light-powered micro/nanorobots have demonstrated remarkable potential for removing microplastics[31]. Au/Ni/$TiO_2$ micromotors have been used to remove microplastic debris in water[32]. Pt-Pd/$Fe_2O_3$ microrobots were able to break the strong covalent bonds in polymer chains through the photo-Fenton reaction[25]. $Fe_3O_4$/$BiVO_4$ microrobots were used to degrade polylactic acid and polycaprolactone in a confined space[33]. Microplastics were enzymatically digested by magnetic field-powered mussel-inspired polydopamine@$Fe_3O_4$/lipase adhesive microrobots as an alternative to the photocatalytic degradation[34]. The adsorptive

bubble separation by bubble-propelled $Fe_2O_3$-$MnO_2$ micromotors has been also identified as an original mechanism for removing microplastics[35]. The main drawback of these microrobots is that the motion occurs at the bottom of the vessel, i.e., in 2D, due to the gravitational force. Since nanoplastics are suspended in water, a strong self-propulsion with six degrees of freedom is required to catch them properly in 3D.

MXenes are promising 2D materials for the fabrication of novel multifunctional microrobots[36]. They exhibit the general formula $M_{n+1}X_nT_x$ ($n = 1, 2, 3$), where M is an early transition metal (Ti, Mo, V), X is carbon and/or nitrogen, and $T_x$ is the surface-terminating functionality ($-O, -F, -OH$)[37]. MXenes are obtained from the selective etching of the A-element layers from a MAX phase, where A is a group IIIA or IVA element (Al, Si)[37]. This process results in "exfoliated" MXene microparticles with high surface area and an accordion-like multi-layered structure. Moreover, they exhibit high conductivity, excellent chemical stability, thermal conductivity, hydrophilicity, surface functionality, and environmental compatibility[38]. As a consequence, they have rapidly emerged in various applications, including water remediation[39,40]. Among the different MXenes, $Ti_3C_2T_x$ is the most studied one. Recently, $Ti_3C_2$ nanoflakes, obtained after the delamination of the exfoliated MXene via ultrasonication, were turned into fuel-free UV-light-driven micromotors thanks to the deposition of a Pt layer on one side and the superficially formed $TiO_2$ in water[41]. These micromotors were able to boost the photocatalytic degradation of nitroaromatic explosives in water despite their low speed and small mean-squared displacement (MSD ~ 5 $\mu m^2$ in 3 s) compared to the other fuel-free light-driven micromotors[25–28,42]. In addition, the delamination involved the destruction of the peculiar multi-layered structure of the MXenes, which is promising for the nanoplastics' trapping.

This work demonstrates the "on-the-fly" capture of nanoplastics in water and their further electrochemical detection through self-propelled light-powered photocatalytic and magnetic MXene-derived oxide multi-layered microrobots. The concept is illustrated in Fig. 1. MXene-derived $\gamma$-$Fe_2O_3$/Pt/$TiO_2$ microrobots were fabricated through a thermal annealing process of exfoliated $Ti_3C_2T_x$ MXene microparticles, followed by the asymmetric deposition of a Pt layer and the subsequent decoration with magnetic $\gamma$-$Fe_2O_3$ nanoparticles. The microrobots showed a negative photogravitactic behavior under UV-light irradiation, resulting in a powerful motion in 3D, and pH-programmable surface charge, which was adequately adjusted to maximize the electrostatic attraction of nanoplastics. SEM and NTA proved the rapid and effective nanoplastics trapping on the microrobots' surface, including the slits between the formerly MXene's multi-layer stacks. Acting as self-motile platforms for fast nanoplastics' preconcentration, the microrobots allowed to detect nanoplastics in water via electrochemical impedance spectroscopy (EIS), using low-cost and miniaturized screen-printed electrodes (SPEs).

## Results and discussion

**Fabrication and characterization of MXene-derived oxide multi-layered microrobots.** MXene-derived $\gamma$-$Fe_2O_3$/Pt/$TiO_2$ microrobots were prepared by thermal annealing of $Ti_3C_2T_x$ MXene, Pt layer deposition, and surface decoration with $\gamma$-$Fe_2O_3$ nanoparticles. Figure 2a reports the scheme of the proposed fabrication procedure. Inspired by the work of Low et al., exfoliated $Ti_3C_2T_x$ MXene microparticles were converted into $TiO_2$ microparticles through a thermal annealing process in air at 550 °C, which has been previously identified as the optimal temperature to prepare $TiO_2$/$Ti_3C_2$ MXene composites[43]. The SEM image in Fig. 2b shows the surface morphology of the pristine $Ti_3C_2T_x$ MXene, which displays the characteristic

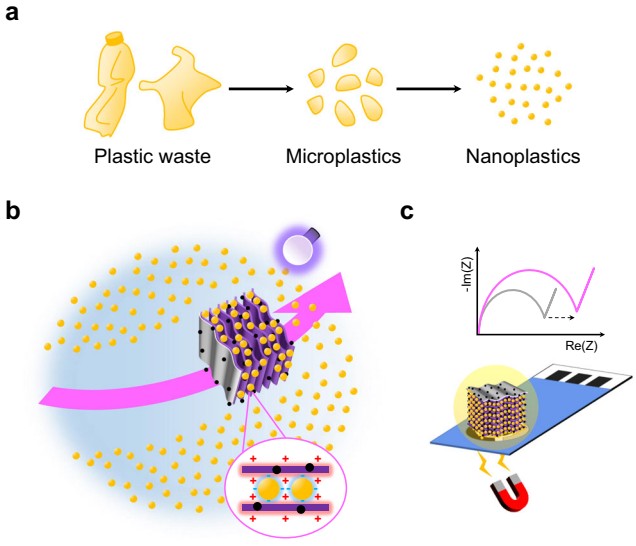

**Fig. 1 Light-powered magnetic MXene-derived γ-Fe₂O₃/Pt/TiO₂ microrobots trap and detect nanoplastics. a** Plastic waste in water fragments into micro- and nanoplastics. **b** Microrobots accelerate the removal of nanoplastics from water by trapping them on their surface, including the slits between multi-layer stacks, due to the combination of a powerful motion and a programmable electrostatic attraction. **c** Acting as self-propelled platforms for preconcentration, microrobots allow nanoplastics' detection by electrochemical impedance spectroscopy (EIS) using low-cost and portable electrodes in an electrolyte solution (yellow shading). The curves represent the impedance real (Re(Z)) and imaginary (-Im(Z)) parts as a function of the frequency (Nyquist plots) of microrobots before (gray) and after capturing nanoplastics (pink).

accordion-like structure consisting of several Ti₃C₂Tₓ multi-layer stacks with a smooth surface. A multi-layer stacks' thickness of ~20 nm is measured from a high-resolution SEM image (Supplementary Fig. 1a). To preserve the MXene multi-layered structure for nanoplastics' trapping, the influence of thermal annealing duration on the morphology of the resulting TiO₂ microparticles was investigated and optimized (Supplementary Fig. 2). After 120 min at 550 °C, the MXene-derived TiO₂ microparticles lose the multi-layered structure due to a significant expansion of the multi-layer stacks, resulting in a continuous and rough surface. By decreasing the duration to 60 min, 30 min, and 0 min, TiO₂ multi-layer stacks' thickness and roughness decrease. Therefore, the optimal MXene-derived TiO₂ microparticles were identified as those kept at 550 °C for 0 min. For this reason, all results presented in the following relate to the 0 min annealing condition. Compared to the Ti₃C₂Tₓ MXene, these microparticles exhibit multi-layer stacks with a rougher surface and a larger thickness of ~50 nm due to the in situ formed TiO₂ nanoparticles (Supplementary Fig. 1b). MXene-derived TiO₂ microparticles were asymmetrically covered with a 50 nm thick Pt layer by sputtering and mixed with γ-Fe₂O₃ nanoparticles for 1 h at room temperature, fabricating light-powered magnetic microrobots. The SEM image in Fig. 2c indicates that the microrobots maintain the multi-layered structure after Pt deposition and γ-Fe₂O₃ nanoparticles' loading. The latter can not be directly visualized on the microrobots due to the high surface roughness of the MXene-derived TiO₂ microparticles and their small size (<50 nm), as indicated by the SEM image of a cluster of γ-Fe₂O₃ nanoparticles (Supplementary Fig. 3). The specific surface area of the Ti₃C₂Tₓ MXene and MXene-derived γ-Fe₂O₃/Pt/TiO₂ microrobots was measured, finding 3.9 and 6.8 m² g⁻¹, respectively. The former

agrees with previous works, reporting values between 0.5 and 9 m² g⁻¹ for multi-layer Ti₃C₂ MXene[43,44]. The larger surface area found for the microrobots is attributed to the thermal annealing process, converting the smooth Ti₃C₂Tₓ surface into TiO₂ nanoparticles[43], Pt layer deposition, and γ-Fe₂O₃ nanoparticles' loading. The preservation of the multi-layered structure is crucial since it avoids sacrificing half of the microparticle's surface upon Pt deposition, as it occurs for smooth spherical particles. It should be noted that layered titanate presents a similar structure to the MXene-derived TiO₂ microparticles, also having the advantage of simpler and cheaper preparation, in principle[45]. Nevertheless, the utilization of commercial exfoliated Ti₃C₂Tₓ MXene and its oxidation into TiO₂ via thermal annealing was considered a more reproducible approach in this work.

X-ray diffraction (XRD) patterns in Fig. 2d prove the transformation of Ti₃C₂Tₓ into anatase TiO₂, which is considered the most photoactive among the different TiO₂ crystal structures[46]. It is worth noting that Low and coworkers found small peaks attributed to Ti₃C₂ in the XRD pattern of the annealed sample and demonstrated a TiO₂/Ti₃C₂ core-shell structure by high-resolution TEM[43]. In this work, those XRD peaks were not found in the XRD pattern of any annealed sample, and thus the residual presence of the Ti₃C₂Tₓ MXene after the thermal annealing process was not claimed and further investigated. Hence, the optimal sample was referred to as an MXene-derived TiO₂ rather than a TiO₂/Ti₃C₂ MXene composite.

Light-absorption spectra of the Ti₃C₂Tₓ MXene and MXene-derived TiO₂ microparticles (thermal annealing condition: 0 min at 550 °C) are reported in Supplementary Fig. 4. The Ti₃C₂Tₓ MXene appears as a black powder absorbing in the visible range from 300 to 700 nm. In comparison, the MXene-derived TiO₂ microparticles appear gray with an absorption edge at 400 nm, as expected for TiO₂, and a weak absorption in the visible range due to unconverted MXene.

X-ray photoelectron spectroscopy (XPS) was performed to study the surface composition and chemical states of the Ti₃C₂Tₓ MXene and MXene-derived TiO₂ microparticles (thermal annealing condition: 0 min at 550 °C). The survey spectra reported in Supplementary Fig. 5 indicate the presence of Ti, C, and O in the two samples, as well as F. The F peak is due to the F⁻ ions of the hydrofluoric acid solution used to etch the MAX phase, and it is more pronounced in the pristine sample. High-resolution XPS spectra of Ti 2p, C 1s, and O 1s for the Ti₃C₂Tₓ MXene (top panels) and MXene-derived TiO₂ microparticles (bottom panels) are shown in Fig. 2e (the binding energy values for all fitted peaks are reported in Supplementary Table 1). Several peaks are identified in the Ti 2p XPS spectrum of the Ti₃C₂Tₓ MXene, consistent with previous reports[47,48]. In contrast, only TiO₂ Ti 2p₃/₂ and Ti 2p₁/₂ peaks are present for the MXene-derived TiO₂ microparticles. A significant difference is observed in the C 1s spectra since the C-Ti-Tₓ peak at 281.7 eV binding energy disappears for the MXene-derived TiO₂ microparticles. Moreover, in the O 1s spectrum of the MXene-derived TiO₂ microparticles, the C-Ti-Oₓ peak is lower than the Ti₃C₂Tₓ MXene, while a pronounced TiO₂ peak is noted. These results confirm the successful conversion of Ti₃C₂Tₓ into TiO₂, in agreement with XRD and optical measurements.

Elemental mapping images of several microparticles were acquired by energy dispersive X-ray spectroscopy (EDX) to verify the deposition of the Pt layer and the surface decoration with γ-Fe₂O₃ nanoparticles (Fig. 2f). These reveal the uniform presence of Ti, O, Pt, and Fe in all microparticles. Besides, the XRD pattern of the final sample exhibited Pt and γ-Fe₂O₃ characteristics peaks (Supplementary Fig. 6). XPS analysis confirmed the presence of Pt⁰ and γ-Fe₂O₃ on the microrobots' surface (Supplementary

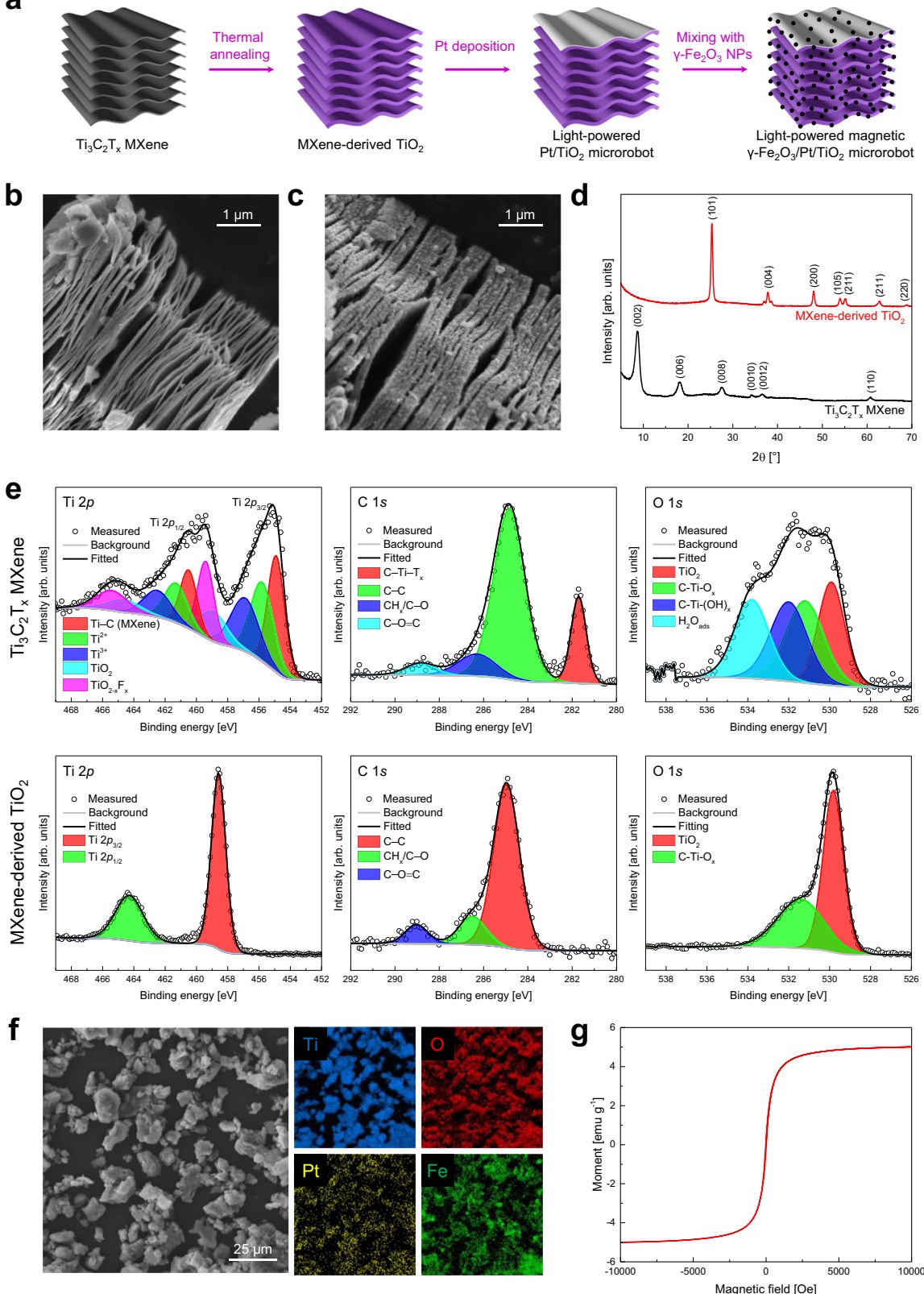

**Fig. 2 Fabrication and characterization of the MXene-derived γ-Fe₂O₃/Pt/TiO₂ microrobots. a** Scheme of the fabrication steps. **b** Ti₃C₂Tₓ MXene microparticle and **c** MXene-derived γ-Fe₂O₃/Pt/TiO₂ microrobot (thermal annealing condition: 0 min at 550 °C) SEM images. **d** XRD patterns of Ti₃C₂Tₓ MXene and MXene-derived TiO₂ microparticles (thermal annealing condition: 0 min at 550 °C). **e** Ti 2p, C 1s, and O 1s high-resolution XPS spectra for the Ti₃C₂Tₓ MXene and MXene-derived TiO₂ microparticles (thermal annealing condition: 0 min at 550 °C). **f** Elemental mapping images for Ti, O, Pt, and Fe and **g** VSM curve of MXene-derived γ-Fe₂O₃/Pt/TiO₂ microrobots (thermal annealing condition: 0 min at 550 °C).

Fig. 7), further proving the successful fabrication of MXene-derived γ-Fe$_2$O$_3$/Pt/TiO$_2$ microrobots. The microrobots were also characterized using a vibrating sample magnetometer (VSM) to study their magnetic properties. The measured magnetization curve in Fig. 2g displays no hysteresis with zero coercivity and remanence, pointing to a superparamagnetic behavior. Supplementary Movie 1 shows the fast magnetic collection of the microrobots in water using a neodymium magnet.

**Microrobots' motion behavior.** Proving the self-propulsion ability of the MXene-derived γ-Fe$_2$O$_3$/Pt/TiO$_2$ microrobots in water under UV-light irradiation is crucial for their further utilization as autonomous platforms to catch and trap nanoplastics. Microrobots' motion was studied in water at pH 7 using an external UV-light source. It was noticed that, without the Pt layer, some MXene-derived TiO$_2$ microparticles could move under UV-light irradiation in the presence of relatively high H$_2$O$_2$ concentrations (>1% H$_2$O$_2$) due to their asymmetric structure. However, their low speed and the required toxic H$_2$O$_2$ made them less attractive than the MXene-derived γ-Fe$_2$O$_3$/Pt/TiO$_2$ microrobots, whose motion did not require fuel or surfactant. Most of the observed microrobots exhibit Brownian motion in dark and autonomous motion under UV-light irradiation on the focal plane (xy plane). Figure 3a reports two frames showing the trajectories of two microrobots after 5 s in dark (left panel) and after 5 s under UV-light irradiation (right panel), while the corresponding video, including also other microrobots, is Supplementary Movie 2. They display a rapid on/off switching of motion with the UV-light, which is reflected in the quick variations of the instantaneous speed vs. time in Fig. 3b. Moreover, a remarkable deceleration is noted within a few s from the beginning of the UV-light irradiation, followed by a plateau. The motion behavior of these microrobots results from the equilibrium between the gravitational force, the buoyancy force, and the driving force of their light-powered self-propulsion, as illustrated in Fig. 3c. The driving force, in combination with the buoyancy force, is not powerful enough to overcome the gravitational force exerted on the microrobots. As a consequence, they can move only at the bottom of the vessel, like for most of the reported semiconductor-based micromotors[16,25]. This typical motion behavior, limited to the xy plane, will be referred to as "2D motion" in this manuscript.

On the contrary, a relevant fraction of the microrobots showed an anomalous motion behavior, which is presented in the frames reported in Fig. 3d for an on/off switching of the UV-light (the corresponding video, including also other microrobots, is Supplementary Movie 3). In the dark, these microrobots display Brownian motion and are indistinguishable from the microrobots moving in 2D. However, under UV-light irradiation, they unfocus in a few s and completely disappear within 5 s, suggesting a movement in the upward direction. This phenomenon is known as negative photogravitaxis, i.e., a particles' migration against gravity when irradiated vertically from the bottom of the substrate, which has been previously observed for other photochemical micromotors[49–51]. By turning off the UV-light, they start focusing and landing at the bottom of the microscope glass slide, being under focus within 5–10 s. A similar behavior has been noticed for visible-light-driven TiO$_2$/Fe$_3$O$_4$/CdS microsubmarines[42]. It is worth remarking that the trajectories reported in Fig. 3d are only the projections on the xy plane of the actual microrobots' trajectories in the 3D space. Figure 3e reports the instantaneous speed of the two microrobots vs. time. While the speed increases quickly by turning on the UV-light, it decreases slowly by turning it off. This is due to the fact that, during microrobots' landing, they show a displacement. Such

passive motion is then translated into a decreasing speed. It was also noted that when turning on the UV-light, most of these microrobots initially undergo a self-orientation, reaching a stand-up position within 100 ms, as shown in Fig. 3f, before starting unfocusing (Supplementary Movie 4). This is ascribed to the powerful driving force exerted on the microrobots, which, together with the buoyancy force, prevails over the gravitational force, inducing their self-orientation and motion with six degrees of freedom, as illustrated in Fig. 3g. This peculiar motion behavior in the xyz space will be referred to as "3D motion".

Microrobots' average speed under UV-light irradiation was estimated from more than 100 different microrobots' trajectories. The measured speed values are reported in the histograms in Fig. 4a for the 2D motion (top panel), i.e., for microrobots moving only on the xy plane, and 3D motion (bottom panel), i.e., for microrobots moving in the xyz space. It is noted that 60% of the analyzed microrobots move in 2D, while 40% move in 3D. The average speed values for the two motion behaviors were obtained by fitting the data with a Gaussian function. Microrobots moving in 2D have an average speed of 9 ± 4 μm s$^{-1}$, while those moving in 3D have a higher speed of 16 ± 8 μm s$^{-1}$. The real speed of the latter is even higher since it was calculated (as it is standard in this field) only for their projections on the xy plane. Nevertheless, the speed of microrobots moving in 2D is still comparable to or higher than the other fuel-free semiconductor-based micromotors[25–28,41,42], which is consistent with the large electrochemical potential difference between MXene-derived TiO$_2$ microparticles and Pt electrodes measured by Tafel experiments in ultra-pure water (Supplementary Fig. 8).

To explain the difference in speed between the microrobots moving in 2D and in 3D, which in turn is responsible for the two observed motion behaviors, some considerations regarding the motion mechanism of the microrobots are necessary. The general motion mechanism of light-powered Pt/TiO$_2$ Janus micromotors is represented in Fig. 4b. TiO$_2$ is an *n*-type semiconductor with a bandgap of 3.2 eV; consequently, it absorbs light in the UV region. Upon exposure to UV-light, electrons are promoted to the TiO$_2$ conduction band, leaving holes in the valence band. The metal/semiconductor Schottky junction at the Pt/TiO$_2$ interfaces facilitates charge separation. Electrons transferred from the TiO$_2$ conduction band to Pt and holes left in TiO$_2$ decompose water according to the reactions reported in Fig. 4b. In particular, the protons (H$^+$) generated at the TiO$_2$ side are consumed at the Pt side to produce H$_2$, establishing an H$^+$ concentration gradient and, thus, a local electric field. This, in turn, drives Pt/TiO$_2$ micromotors' motion via a self-electrophoretic mechanism with TiO$_2$ as the forward side[52]. In this context, the Pt coating plays a key role. Pt deposition can be affected by the morphology of the underlying semiconductor microparticle. In fact, for UV-light-powered Pt/ZnO Janus micromotors, it has been demonstrated that a smooth ZnO microparticle results in a continuous Pt coating and the fuel-free motion ability, while a rough ZnO microparticle leads to a discontinuous Pt coating which requires H$_2$O$_2$ fuel for the autonomous propulsion[27]. Similarly, smooth TiO$_2$ microparticles coated by a continuous Pt layer exhibited a higher speed under UV-light-irradiation in H$_2$O$_2$ than rough ones[53]. Moreover, it has been shown that transforming a continuous Pt film on SiO$_2$ microspheres into discrete Pt nanoparticles via thermal annealing causes an ~80% decrease in their self-electrophoretic speed due to the lower electric field or fluid flow generated by the discontinuous Pt coating[54].

Contrarily to conventional semiconductor microparticles, the MXene-derived TiO$_2$ microparticles present an intrinsic asymmetry owing to the formerly MXene multi-layered structure. By ideally assuming these microparticles as cubes/parallelepipeds formed by several adjacent layers, it is observed that four faces out

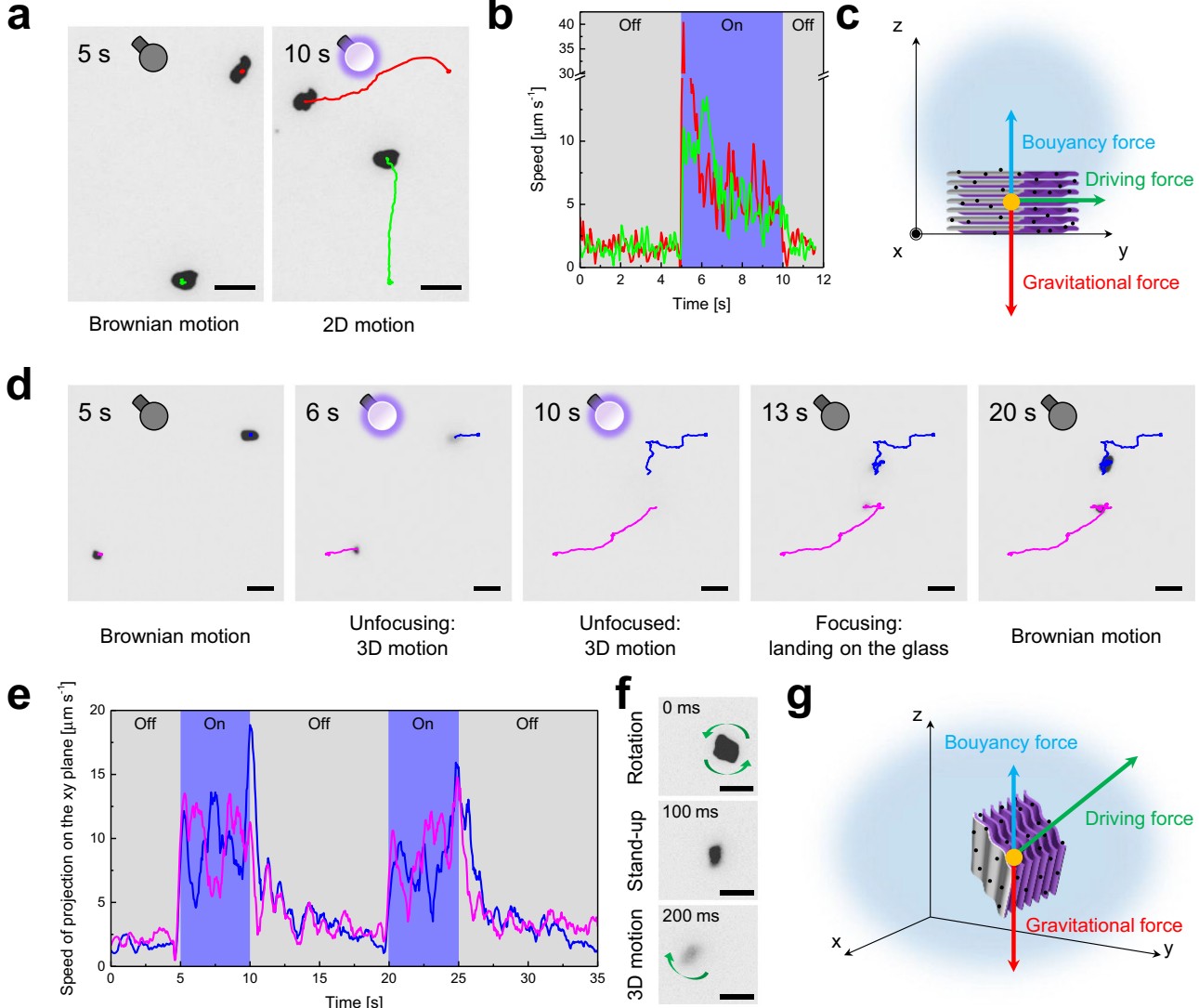

**Fig. 3 Motion behavior of the MXene-derived γ-Fe₂O₃/Pt/TiO₂ microrobots in water under light irradiation. a** Representative trajectories of the MXene-derived $\gamma$-Fe$_2$O$_3$/Pt/TiO$_2$ microrobots (thermal annealing condition: 0 min at 550 °C) moving on the xy plane (2D motion) in fuel-free water at pH 7 in the absence and presence of UV-light irradiation, **b** their corresponding instantaneous speed vs. time, and **c** force analysis. **d** Representative trajectories of the MXene-derived $\gamma$-Fe$_2$O$_3$/Pt/TiO$_2$ microrobots (thermal annealing condition: 0 min at 550 °C) moving in the xyz space (3D motion) in fuel-free water at pH 7 in the absence and presence of UV-light irradiation, **e** their corresponding instantaneous speed vs. time (trajectories and speeds are the projections of the actual microrobots' trajectories and speeds on the xy plane), **f** representative self-orientation by switching on the UV-light irradiation, and **g** force analysis. Scale bars are 10 μm.

of six are layered and discontinuous. At the same time, the remaining two are flat and continuous. Pt deposition by sputtering technique is conducted on substrates placed in front of the Pt target. These substrates consist of microscope glass slides on which a suspension of MXene-derived TiO₂ microparticles is dropped and dried overnight to form a monolayer of particles. During this preliminary preparation step, it can be assumed that the particles randomly deposit on the glass slides. So, in the ideal case, there is a probability of ~67% and ~33% to have one of the multi-layered or flat faces like the one exposed to the Pt target, as depicted in Fig. 4c. These probabilities are in good agreement with the relative frequency of 2D and 3D moving microrobots (60 and 40%, respectively), suggesting that a continuous Pt deposition on the flat side of the MXene-derived TiO₂ microparticle may be the origin of the higher speed and stronger propulsive force of the resulting microrobots. This powerful driving force, combined with the buoyancy, overcomes the gravitational force and unlocks

the microrobots' movement in the upward direction, leading to their 3D motion ability. To corroborate this conclusion, a 2D simulation of the H₂ production from the Pt side by consuming the protons generated under UV-light irradiation at the TiO₂ side was performed for the two configurations illustrated in Fig. 4c. The simulated H₂ concentration spatial distribution after 0.1 s irradiation is shown in Fig. 4d. The continuous Pt coating on the flat side of the MXene-derived TiO₂ microparticle produces more H₂ compared to the discontinuous Pt coating on the multi-layered side, reflecting a larger H⁺ concentration gradient and, so, a higher speed and 3D motion ability. This conclusion is consistent with the previously reported stronger propulsion for microrobots coated by a continuous Pt layer[27,53,54].

**Nanoplastics' capture**. As representatives for nanoplastic debris, carboxylated polystyrene nanoparticles with a diameter of 50 nm

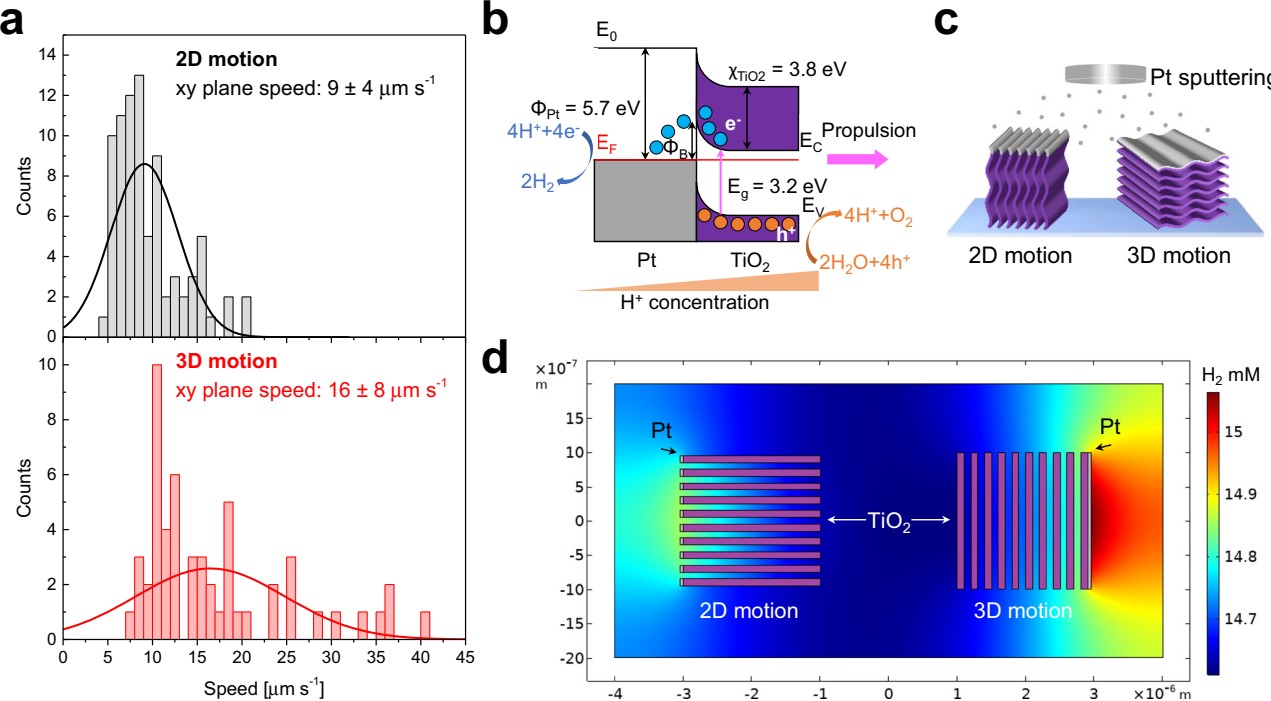

**Fig. 4 Analysis of the MXene-derived γ-Fe₂O₃/Pt/TiO₂ microrobots' 2D and 3D motion. a** Speed distributions of the MXene-derived γ-Fe₂O₃/Pt/
TiO₂ microrobots (thermal annealing condition: 0 min at 550 °C) showing 2D and 3D motion in water at pH 7 under UV-light irradiation. **b** Scheme of the
general motion mechanism of Pt/TiO₂ microrobots ($E_0$ is the vacuum level, $E_F$ is the Fermi level, $\Phi_{Pt}$ is Pt work function, $\Phi_B$ is the Schottky barrier height,
$E_C$, $E_V$, $E_g$, and $\chi_{TiO2}$ are TiO₂ conduction and valence band levels, optical bandgap, and electron affinity). **c** Scheme of the two MXene-derived γ-Fe₂O₃/Pt/
TiO₂ microrobots' configurations originated from the MXene-derived TiO₂ microparticles' multi-layered structure and orientation with respect to the Pt
target during Pt deposition. **d** Numerical simulation of the H₂ concentration spatial distribution produced in 0.1 s for the cross-section of the two
configurations.

were used. The concentration of nanoplastics in water samples
was determined by NTA. This method allows measuring the size
distribution and concentration of nanoparticles in environmental,
biological, and food samples[55]. In particular, NTA has been used
to estimate the concentration of nanoplastics in water samples
and the release of nanoplastics as a result of (micro)plastics'
degradation[11]. In addition, it has been employed to measure the
diffusion coefficient of autonomous nanomotors[56,57]. The work-
ing principle of NTA is schematically illustrated in Fig. 5a. A laser
beam passes through the nanoplastics' suspension in the sample's
chamber. The light scattered by the nanoplastics is observed and
recorded using a microscope equipped with a camera. A software
tracks the nanoplastics' Brownian motion at a fixed temperature
and, using the Stokes-Einstein equation, calculates their hydro-
dynamic diameters.

Figure 5b shows the average size distribution of the
nanoplastics' suspension after a dilution by a factor $5 \times 10^4$,
obtained through NTA. An average size of 50 nm is noted, as
expected. The area of the size distribution yields the concentra-
tion of nanoplastics, which is ~$6 \times 10^9$ nanoplastics ml⁻¹ after the
dilution. Based on the information provided by the supplier of the
carboxylated polystyrene nanoparticles' suspension and polystyr-
ene density (1.05 g cm⁻³), nanoplastics' concentration before the
dilution should be $1.5 \times 10^{14}$ nanoplastics ml⁻¹ (Supplementary
Discussion 1). After the dilution, the nanoplastics' concentration
should be $3 \times 10^9$ nanoplastics ml⁻¹. This value is in good
agreement with the nanoplastics' concentration measured
through NTA, demonstrating the reliability of the proposed
methodology. Before the nanoplastics' capture experiments with
the microrobots, the optimal nanoplastics' concentration range
for NTA was determined by serial dilution. Frames from the

videos recorded for the various nanoplastics' suspensions are
shown in Fig. 5c (the videos are reported in Supplementary
Movie 5). Nanoplastics are visualized as bright spots on a dark
background. It is noticed that the number of the spots decreases
by increasing the dilution factor (DF). The corresponding
calibration curve is displayed in Fig. 5d. The nanoplastics'
concentration (~$6 \times 10^9 \div 2 \times 10^7$ nanoplastics ml⁻¹) is linear
over a wide range of dilutions ($5 \times 10^4 \div 1 \times 10^7$).

The proposed nanoplastics' remediation strategy is represented
in Fig. 6a. The MXene-derived γ-Fe₂O₃/Pt/TiO₂ microrobots
accelerate nanoplastics' trapping thanks to the powerful self-
propulsion ability under light irradiation and an engineered
electrostatic attraction. Then, the microrobots and the trapped
nanoplastics are collected magnetically and separated from the
treated water. The electrostatic attraction has already been
addressed as an effective strategy for capturing polymer chains
and microplastics[15,25]. Therefore, the Zeta potentials of the
microrobots and nanoplastics were measured at different pH to
maximize their electrostatic attraction (Fig. 6b). The nanoplastics
used in this work are negatively charged due to the surface
carboxylic groups. At pH 3, their Zeta potential is −60 mV.
Contrarily, the microrobots are only slightly positive at pH 7,
displaying a Zeta potential of +12 mV. This increases up to
+43 mV at pH 3. Therefore, all nanoplastics capture experiments
have been conducted in pH 3 water. Microrobots' motility in
water at pH 3 was studied. About 13% of the recorded
microrobots show the 3D motion, although at a lower average
speed than pH 7 (Supplementary Fig. 9). Also, the speed of the
2D motion decreases significantly. This behavior is explained by
the higher ionic strength of water at pH 3 or the lower
microrobots' photochemical activity[58]. In addition, the attraction

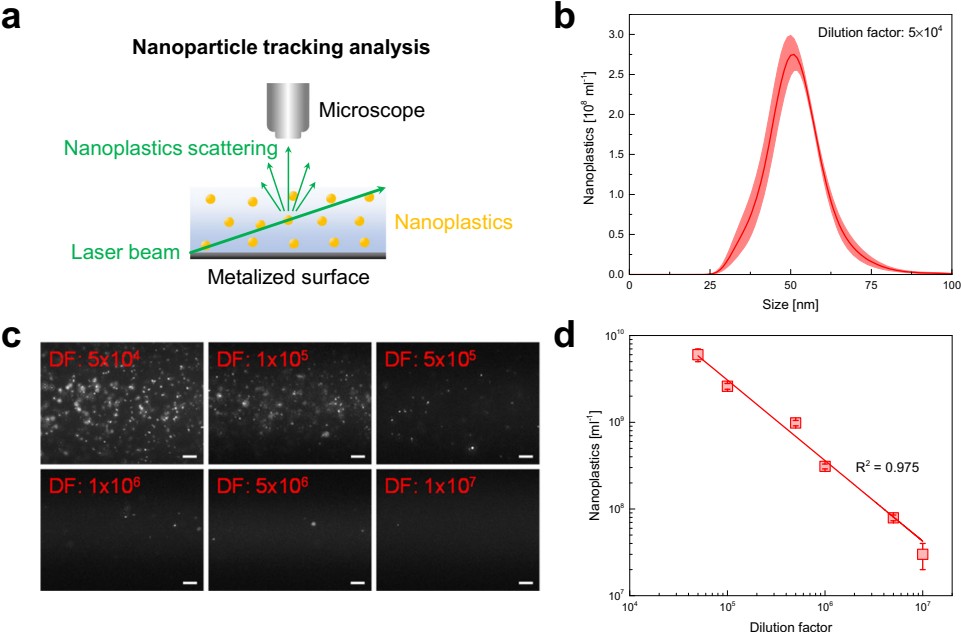

**Fig. 5 Nanoparticle tracking analysis (NTA). a** Scheme of the NTA working principle. **b** Size distribution of the nanoplastics' suspension (carboxylated polystyrene nanoparticles in water) after the dilution by a factor $5 \times 10^4$ measured by NTA. The shaded region replaces the discrete error bars calculated as the standard error for $n = 3$ independent replicates. **c** Frames of the videos of nanoplastics' suspensions after serial dilution and **d** the corresponding calibration curve showing the linear range for NTA. Scale bars are 10 μm. Error bars represent the standard deviation, $n = 3$ independent replicates.

between the positively charged microrobots and the negatively charged glass slide may contribute to slowing down the microrobots, which more often are stuck to the substrate.

Figure 6c reports the SEM images of a microrobot after exposure to the nanoplastics' suspension before dilution for 1 min under UV-light irradiation. Nanoplastics are noticed on the microrobot's surface, including the slits between the formerly MXene's multi-layer stacks, demonstrating the efficacy of the multi-layered design and electrostatic trapping. A quantitative study of nanoplastics' capture for different treatment durations (1, 3, and 5 min) was conducted through the NTA. To exploit the entire linear range, the initial nanoplastics' concentration was set to ~$6 \times 10^9$ nanoplastics ml$^{-1}$. γ-Fe$_2$O$_3$/Ti$_3$C$_2$T$_x$ MXene microparticles served as a reference to better evaluate the contribution of microrobots' self-propulsion to the nanoplastics' capture. Figure 6d shows frames from the videos of the nanoplastics' suspensions after the treatments with MXene and microrobots (the videos are reported in Supplementary Movie 6). At the same time, the measured nanoplastics' concentration values are shown in Fig. 6e. Most nanoplastics are still present even after the more prolonged treatment with the MXene microparticles. In particular, after 5 min-long treatment, the concentration of nanoplastics decreased from $(6 \pm 1) \times 10^9$ to $(3 \pm 1) \times 10^9$ nanoplastics ml$^{-1}$, which corresponds to a nanoplastics' capture efficiency of 50%. It is worth noting that the scattering of the nanoplastics increased after the treatment, which is reflected in the brighter spots in Fig. 6d for 1, 3, and 5 min. On the contrary, the microrobots catch 97% of the nanoplastics within 1 min, leaving only $(4.4 \pm 0.6) \times 10^7$ nanoplastics ml$^{-1}$ in the water sample after 5 min. In fact, the MXene microparticles land on the bottom of the vessel. Instead, the microrobots can capture many more nanoplastics due to their active motion. Control experiments using the microrobots without UV-light irradiation, i.e., in the static condition, revealed a trend similar to the MXene microparticles (Supplementary Fig. 10), further confirming the crucial contribution of microrobots' self-propulsion to the nanoplastics' capture process.

The microrobots' removal capacity $q_t$ [mg g$^{-1}$] was calculated and plotted in Supplementary Fig. 11 to allow a straightforward comparison with "conventional" materials tested under similar conditions in recent studies. The microrobots reached a $q_t$ of $0.5 \pm 0.1$ mg g$^{-1}$ within 1 min treatment. This value is close to that reported for other materials, such as granular activated carbon (~0.3-0.7 mg g$^{-1}$ after 15 min)[59], cellulose fibers (0.8–0.86 mg g$^{-1}$ after 5–120 min)[60], and untreated coffee grains (>2 mg g$^{-1}$ after 5 min)[61], despite having a larger surface area and being utilized under external agitation.

The microrobots' reusability has been tested by several nanoplastics' capture and release cycles (Supplementary Fig. 12). It was found that they could be reused many times as their capture ability decreased slightly after each run.

**Nanoplastics' detection.** The microrobots' rapid nanoplastics collection ability inspired their use as self-propelled platforms for fast nanoplastics' preconcentration and further detection through the EIS technique as a cheap, fast, and portable alternative to NTA. The measurement involves a conventional three-electrode electrochemical cell comprising working, reference, and counter electrodes connected to a potentiostat and a redox probe in the electrolyte solution. A small-amplitude sinusoidal voltage is applied to the working electrode at several frequencies. The fit of the EIS spectrum based on an equivalent circuit model allows the determination of the circuit parameters. By monitoring these parameters, EIS can be used to sensitively detect impedance variations at the electrolyte/electrode interface caused by the adsorption of molecules or particles[62].

Here, the three-electrode setup has been replaced with commercial SPEs, which have the advantage of miniaturization and portability, low cost, and low sample volume[63]. To demonstrate the importance of nanoplastics' preconcentration for sensing, the EIS spectra of an SPE in the absence and presence of nanoplastics (~$10^{12}$ nanoplastics ml$^{-1}$) were first measured using Fe(CN)$_6^{4-/3-}$ as the redox probe. Figure 7a reports the obtained Nyquist plots, showing the impedance real and

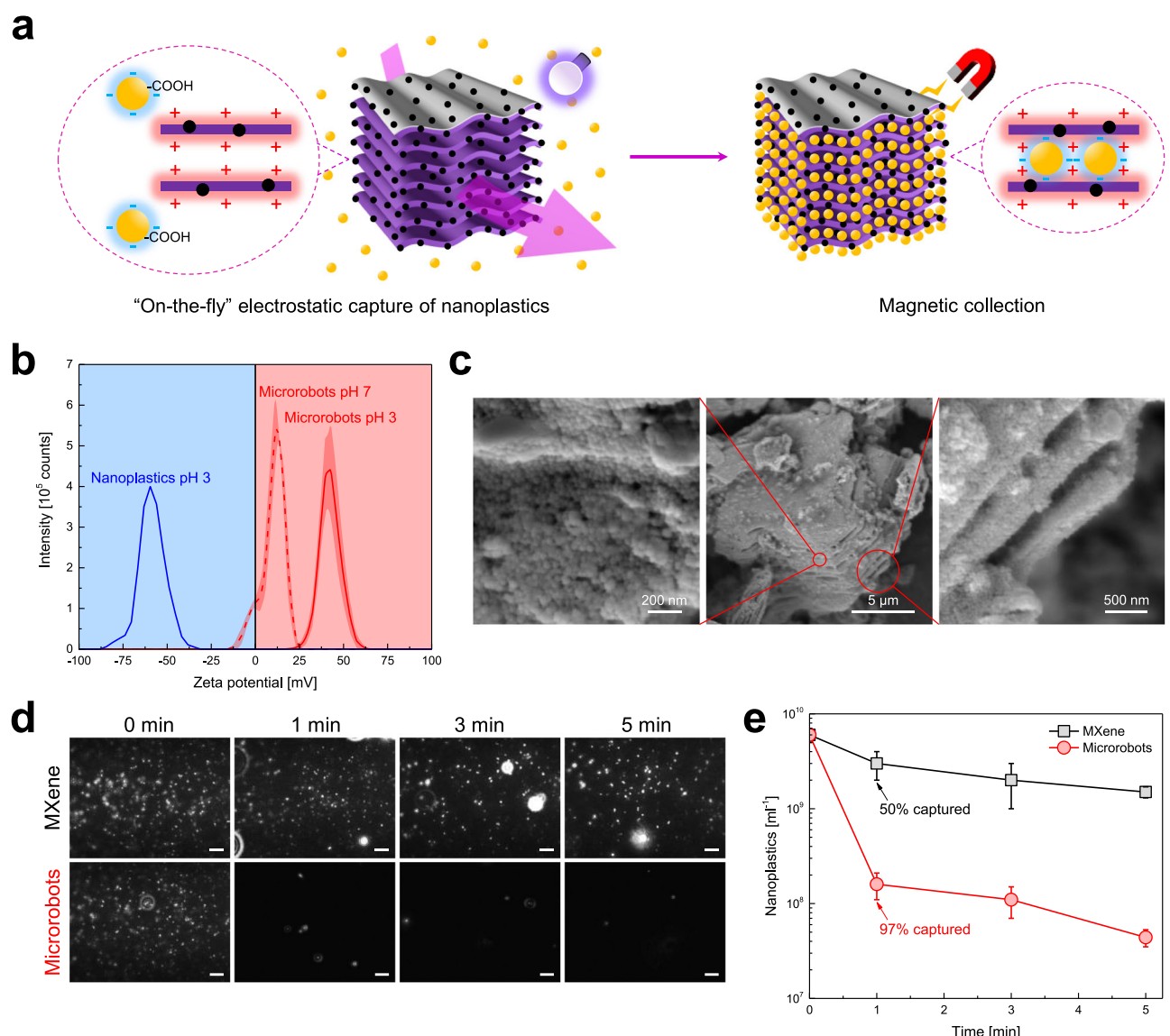

**Fig. 6 Nanoplastics' capture by MXene-derived γ-Fe₂O₃/Pt/TiO₂ microrobots. a** Scheme of the nanoplastics' remediation strategy: nanoplastics are electrostatically collected by the self-propelled light-powered MXene-derived γ-Fe₂O₃/Pt/TiO₂ microrobots and magnetically removed from the treated water using a magnetic field. **b** Zeta potential of the MXene-derived γ-Fe₂O₃/Pt/TiO₂ microrobots (thermal annealing condition: 0 min at 550 °C) and nanoplastics in water at various pH. The shaded regions replace the discrete error bars calculated as the standard error for $n = 3$ independent replicates. **c** SEM images of an MXene-derived γ-Fe₂O₃/Pt/TiO₂ microrobot (thermal annealing condition: 0 min at 550 °C) after exposure to the nanoplastics suspension (capture experiment conditions: 0.75 mg ml⁻¹ microrobots, 1.5 × 10¹⁴ nanoplastics ml⁻¹, water at pH 3, UV-light irradiation for 1 min, microrobots' collection using a neodymium magnet). **d** Frames of the videos of nanoplastics' suspensions after the treatments with the γ-Fe₂O₃/Ti₃C₂ₓ MXene microparticles and MXene-derived γ-Fe₂O₃/Pt/TiO₂ microrobots for different durations (1, 3, and 5 min) under UV-light irradiation (capture experiments conditions: 0.75 mg ml⁻¹ sample, 6 × 10⁹ nanoplastics ml⁻¹, water at pH 3, samples' collection using a neodymium magnet), and **e** the corresponding nanoplastics' concentration values obtained by NTA. Scale bars are 10 μm. Error bars represent the standard deviation, $n = 3$ independent replicates.

imaginary parts as a function of the frequency (the corresponding Bode plots, showing impedance modulus and phase as a function of the frequency, are reported in Supplementary Fig. 13). The two curves are almost overlapped and display a small semicircle arc in the high-frequency region (left) and a straight line in the low-frequency region (right). The minor difference between these curves suggests that a bare SPE is insufficient to unambiguously discriminate a nanoplastics-contaminated water sample from a pure one.

An aqueous suspension of the microrobots with captured nanoplastics was dropped onto an SPE to perform the EIS measurement. The SPE was equipped with a magnet on the backside to accelerate the landing of the microrobots and stick them on the working electrode area during the replacement of water with the electrolyte. The same procedure was executed for a suspension of microrobots without nanoplastics, which served as a reference. The Nyquist plots of microrobots with and without captured nanoplastics are also presented in Fig. 7a. A large semicircle is noted in both cases, while the straight line is just hinted. Nevertheless, the diameter of the semicircle in the presence of the nanoplastics is significantly greater. Such variation agrees with a previous report on the increased resistance of conductive membranes after the deposition of latex beads measured by EIS, and it is promising for nanoplastics' detection[64].

Figure 7a shows the fit of the EIS spectra according to the equivalent circuit model depicted in Fig. 7b. This consists of the

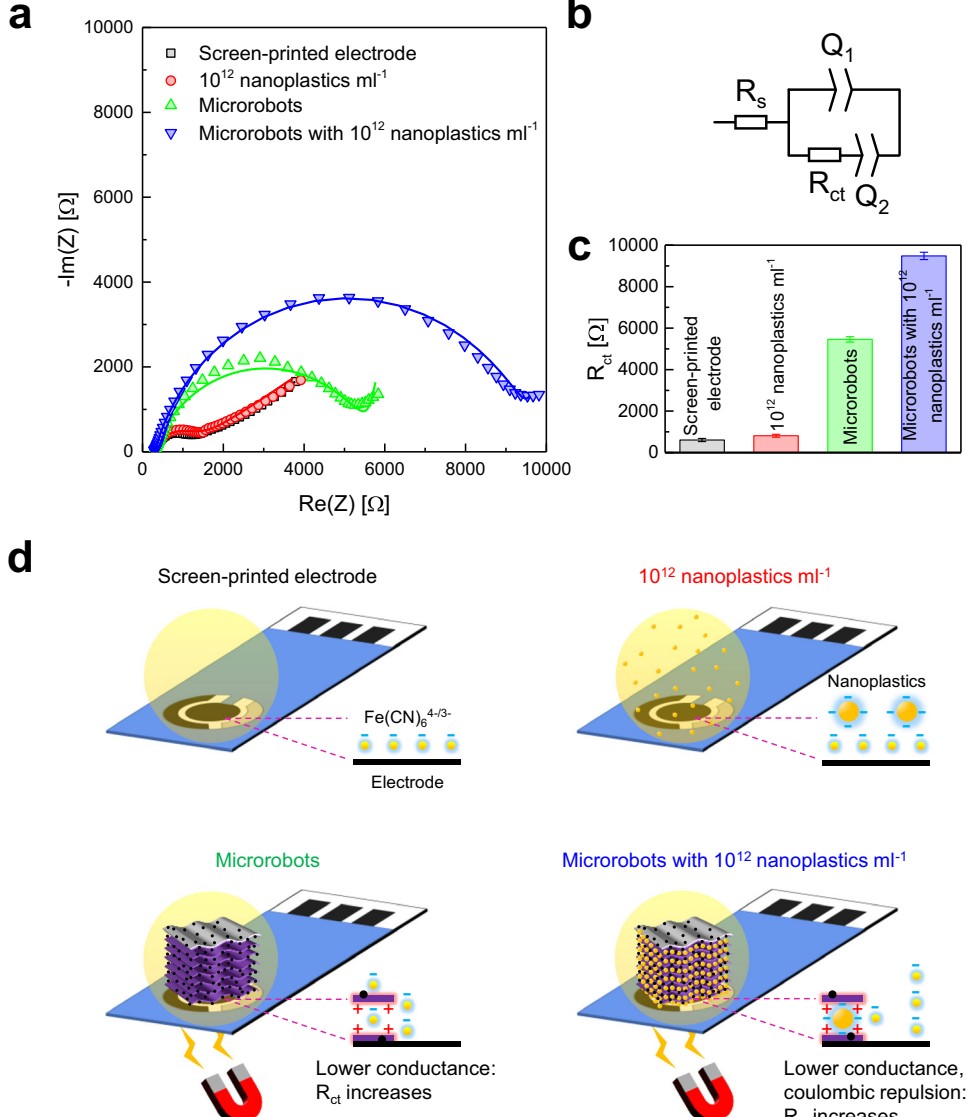

**Fig. 7 Nanoplastics' detection by electrochemical impedance spectroscopy (EIS) technique after the preconcentration with the MXene-derived γ-Fe₂O₃/Pt/TiO₂ microrobots. a** Nyquist plots showing the impedance real (Re(Z)) and imaginary (-Im(Z)) parts as a function of the frequency for a bare screen-printed electrode (SPE), an SPE exposed to ~$10^{12}$ nanoplastics ml⁻¹ suspension, an SPE after MXene-derived γ-Fe₂O₃/Pt/TiO₂ microrobots' (thermal annealing condition: 0 min at 550 °C) loading, and an SPE after loading microrobots with captured nanoplastics (capture experiment conditions: 0.75 mg ml⁻¹ microrobots, ~$10^{12}$ nanoplastics ml⁻¹, water at pH 3, UV-light irradiation for 1 min, microrobots' collection using a neodymium magnet). The EIS measurements were performed in a 10 mM Fe(CN)₆⁴⁻/³⁻ aqueous solution at 0 V vs. ref. with a 10 mV superimposed sinusoidal root-mean-square voltage in the frequency range $10^5 \div 10^0$ Hz. The lines represent fits to the data. **b** Equivalent circuit model for the Nyquist plots fit: the electrolyte solution resistance (Rₛ) is in series with the double-layer capacitance of the working electrode described by a constant phase element (Q₁), which is in parallel with the charge transfer resistance at the electrolyte/electrode interface (Rct) and a second constant phase element (Q₂). **c** Rct values obtained from the Nyquist plots fit. Error bars represent the fitting error. **d** Schemes of the different EIS measurements and electrolyte/electrode interfaces representing a bare SPE, an SPE exposed to a nanoplastics' suspension, an SPE where microrobots are loaded on the working electrode, and an SPE where microrobots with captured nanoplastics are loaded on the working electrode in a Fe(CN)₆⁴⁻/³⁻ solution.

electrolyte solution resistance (Rₛ) in series with the double-layer capacitance of the working electrode, described through a constant phase element (Q₁) to take into account the nonideality of the capacitor, in parallel with the charge transfer resistance at the electrolyte/electrode interface (Rct) and a second constant phase element (Q₂)[65]. The calculated circuit parameters are reported in Supplementary Table 2. Among all parameters, the most significant variations are observed for Rct, which corresponds to the diameter of the semicircle in the Nyquist plot. Figure 7c displays the calculated Rct values. A negligible increase in Rct is observed for the SPE upon exposure to the nanoplastics'

suspension. The Rct significantly rises after microrobots loading onto the SPE and almost doubles for microrobots with nanoplastics. This observation is consistent with transient photocurrent response measurements, showing a microrobots' photocurrent decrease after capturing nanoplastics (Supplementary Fig. 14).

The scheme in Fig. 7d allows to better understand the impedimetric behaviors of the samples. Since the nanoplastics are suspended in the solution, the electrolyte/SPE interface remains unchanged. Hence, there is only a tiny difference in the impedance of the bare SPE with and without nanoplastics. The

presence of the microrobots leads to a notable increase of the SPE's $R_{ct}$ due to the low conductivity of the semiconducting TiO$_2$, which represents the main constituent of the microrobots[66]. This behavior has also been observed after TiO$_2$ loading on FTO electrodes[67]. The $R_{ct}$ further rises due to the lower microrobots' conductance in consequence of the inclusion of a large quantity of non-conducting polystyrene nanoparticles[68]. Additionally, nano-plastics' capture results in the neutralization of microrobots' positive surface charge and, thus, the Coulomb repulsion of the redox probe ions (Fe(CN)$_6^{4-/3-}$), as demonstrated by the microrobots' Zeta potential decrease ($+43 \rightarrow -1$ eV) after nanoplastics' capture (Supplementary Fig. 15). This mechanism, similar to the one at the base of DNA hybridization sensors, contributed to the $R_{ct}$ increase[62].

The method's reliability has been verified by measuring the impedance of microrobots exposed to different nanoplastics' concentrations ($\sim10^6$ and $10^{14}$ nanoplastics ml$^{-1}$). Supplementary Fig. 16 shows the recorded Nyquist and Bode plots, whose fitting parameters are reported in Supplementary Table 2. The $R_{ct}$ for the bare microrobots is similar to microrobots with $10^6$ nanoplastics ml$^{-1}$, suggesting that this low concentration can not produce a detectable impedance variation. On the other hand, a remarkable increase of the $R_{ct}$ ($\sim18000\ \Omega$) was obtained for the microrobots with $10^{14}$ nanoplastics ml$^{-1}$, corroborating the consistency of the proposed sensing strategy.

On these bases, the microrobots quickly accumulate nanoplastics in water and allow determining their presence at concentrations higher than $10^6$ nanoplastics ml$^{-1}$ through impedance measurements. The rapidity and efficiency of nanoplastics' preconcentration and EIS measurement, in combination with the low cost and compactness of the SPEs, hold promise for the "on-site" nanoplastics' sensing in wastewater.

In conclusion, an efficient strategy for removing and detecting nanoplastics in water is presented based on innovative self-propelled light-powered magnetic MXene-derived microrobots. $\gamma$-Fe$_2$O$_3$/Pt/TiO$_2$ microrobots were prepared by simple thermal annealing in air of exfoliated Ti$_3$C$_2$T$_x$ MXene microparticles, followed by the deposition of a Pt layer and the surface decoration with magnetic $\gamma$-Fe$_2$O$_3$ nanoparticles. The duration of the annealing process was optimized to preserve the characteristic multi-layered structure of MXenes. The microrobots showed a powerful motion in fuel-free water under UV-light irradiation with quick on/off switching. The Pt deposition on the flat sides of the MXene-derived TiO$_2$ microparticles rather than on the multi-layered sides was identified as the reason for the observed higher speed and negative photogravitactic behavior, resulting in the 3D movement of the microrobots, in agreement with numerical simulations. This result proves that a continuous Pt coating generates a stronger micro/nanorobots' propulsion than a discontinuous one. The ability of microrobots to trap nanoplastics was evaluated through nanoplastics' capture experiments under UV-light irradiation using carboxylated polystyrene beads (50 nm in size) as a model for nanoplastics in water. SEM analysis confirmed the trapping of nanoplastics on the microrobots' surface, including the slits between the formerly MXene's multi-layer stacks. Compared to the static MXene microparticles, the microrobots demonstrated a superior capture efficiency (97% vs. 50% in 1 min), as confirmed by NTA, due to their fast active motion and tailored Zeta potential to attract the oppositely charged nanoplastics. The superparamagnetic behavior of the microrobots allowed their facile collection from the treated water. Owing to their rapid and excellent nanoplastics' capture capability, the microrobots were employed as self-motile plat-forms for the preconcentration and successive electrochemical detection of nanoplastics using low-cost SPEs by the EIS technique. Transferring the self-propulsion, photocatalytic,

magnetic, and pH-programmable surface charge properties of the microrobots to larger surface area materials would signifi-cantly enhance the nanoplastics' capture efficiency and, in principle, reduce the preconcentration time and improve the electrochemical sensor's sensitivity. This study lays the basis for the "on-site" screening of nanoplastics in wastewater. The multi-layered design of the microrobots is also promising for the removal and subsequent photocatalytic degradation of other organic pollutants in water.

## Methods

**Materials and reagents.** Exfoliated Ti$_3$C$_2$T$_x$ MXene was purchased from Laizhou Kai Kai Ceramic Materials Co. Ltd (Hong Kong S.A.R.). Calcium chloride (CaCl$_2$, ≥98%), iron sulfate heptahydrate (FeSO$_4$•7H$_2$O, ACS reagent ≥99%), ammonium hydroxide (NH$_4$OH, 25% NH$_3$ in H$_2$O), potassium hexacyanoferrate(III) (K$_3$Fe(CN)$_6$, ACS reagent ≥99%), and indium tin oxide-coated glass slides (ITO, $75 \times 25 \times 1.1$ mm$^3$, 8–12 Ω sq$^{-1}$ surface resistivity) were purchased from Sigma-Aldrich (Merck, Germany). Iron chloride hexahydrate (FeCl$_3$•6H$_2$O, 97%) was purchased from Alfa Aesar (Thermo Fisher Scientific, US). Pt target was purchased from Neyco (France). Carboxylated polystyrene nanoparticles (1% solid content in 5 ml water, corresponding to 10 mg ml$^{-1}$, 50 nm in size) were purchased from Degradex (Phosphorex, US). ET077-40 SPEs ($50 \times 13$ mm$^2$, 3 mm diameter disk working electrode, graphitic carbon powder working and auxiliary electrodes, Ag/AgCl pellet reference electrode) were purchased from Zensor (Taiwan).

**Fabrication of MXene-derived Pt/TiO$_2$ microrobots.** Exfoliated Ti$_3$C$_2$T$_x$ MXene microparticles were transformed into TiO$_2$ microparticles by a thermal annealing process conducted in a tubular furnace in air with a heating rate of 10 °C min$^{-1}$ up to 550 °C. To preserve the accordion-like structure of the MXene, the influence of the thermal annealing process duration (0, 30, 60, and 120 min) on the morpho-logical and structural properties of the resulting TiO$_2$ microparticles was investi-gated. To fabricate MXene-derived Pt/TiO$_2$ microrobots, a suspension of the optimal (thermal annealing condition: 0 min at 550 °C) MXene-derived TiO$_2$ microparticles (5 mg ml$^{-1}$) was prepared using ultra-pure water (18 MΩ cm), dropped on glass slides, and dried overnight. Then, a Pt layer (50 nm) was asymmetrically deposited on the microparticles using a Leica EM ACE600 high vacuum sputter coater. The obtained MXene-derived Pt/TiO$_2$ microrobots were detached from the glass slides through a scalpel.

**Synthesis of $\gamma$-Fe$_2$O$_3$ nanoparticles for microrobots' surface decoration.** To synthesize magnetic $\gamma$-Fe$_2$O$_3$ nanoparticles, an aqueous solution containing FeSO$_4$•7H$_2$O and FeCl$_3$•6H$_2$O (molar ratio 1:1) was stirred for 15 min at 60 °C. NH$_4$OH was added dropwise until pH 11 was reached and further stirred for 15 min at 60 °C. The as-synthesized $\gamma$-Fe$_2$O$_3$ nanoparticles were washed two times in ultra-pure water and ethanol, and dried overnight.

The decoration of the microrobots with $\gamma$-Fe$_2$O$_3$ nanoparticles was carried out by adding 1 mg of the nanoparticles to a 2 ml solution of ultra-pure water and ethanol (volume ratio 1:1), followed by sonication for 30 s. 10 mg microrobots were added to the solution (microrobots to nanoparticles ratio 1:10) and stirred for 1 h at room temperature. Finally, MXene-derived $\gamma$-Fe$_2$O$_3$/Pt/TiO$_2$ microrobots were magnetically collected using a neodymium magnet, washed two times in ultra-pure water and ethanol, and dried overnight. The same procedure was used to prepare $\gamma$-Fe$_2$O$_3$/Ti$_3$C$_2$T$_x$ MXene microparticles used as a reference in nanoplastics' capture experiments.

**Characterization techniques.** Surface morphology and elemental composition were characterized by a TESCAN MIRA3 XMU SEM equipped with an Oxford Instruments energy dispersive X-ray (EDX) detector. The Brunauer–Emmett–Teller (BET) specific surface area was evaluated from the adsorption data registered through a 3P micro 300 instrument in the relative pressure (p/p$_0$) range of 0.1 ÷ 0.3. Before BET measurements, samples were degassed at 300 °C overnight[44]. The crystalline structure was determined by XRD in parallel beam mode using a Rigaku SmartLab 9 kW diffractometer equipped with a high-brightness Cu K$_\alpha$ rotating anode X-ray tube operated at 45 kV and 150 mA. Surface chemical composition was studied by XPS using a Kratos Analytical Axis Supra instrument with a monochromatic Al K$_\alpha$ (1486.7 eV) excitation source. All spectra were calibrated to the adventitious C 1$s$ peak at 284.8 eV and fitted using CasaXPS software. The magnetic hysteresis loop was measured using a Quantum Design VersaLab cryogen-free VSM at 300 K for an applied magnetic field ranging from −10 to 10 kOe at steps of 10 Oe s$^{-1}$. Light-absorption spectra were measured using a Jasco V-750 UV–Visible spectrophotometer equipped with an integrating sphere. Zeta potential measurements were performed in water at pH 7 and 3 using a Malvern Panalytical Zetasizer Ultra instrument.

**Microrobots' motion behavior.** The light-powered motion behavior of micro-robots in water was observed and recorded using a Nikon ECLIPSE TS2R inverted microscope coupled to a BASLER acA1920-155uc digital camera. All experiments were carried out in the absence of any surfactant. A 365 nm UV LED (Cool LED

pE-100) operated at 1600 mW cm$^{-2}$ was used as the light source to power the microrobots. Videos were recorded at 20 fps and analyzed through NIS Elements Advanced Research and Fiji software to obtain microrobots' trajectories and measure their speed.

**Tafel and transient photocurrent response measurements**. Tafel measurements were carried out using a customized photoelectrochemical setup with three LZ4-04UV00 365 nm UV LED (LedEngin Inc.) in the two-electrode configuration with TiO$_2$ electrode or Pt electrode as working electrodes and an Ag/AgCl electrode (1 M KCl) as both reference and counter electrode. The TiO$_2$ working electrode was prepared by dropping 100 μl of an aqueous suspension of optimal (thermal annealing condition: 0 min at 550 °C) MXene-derived TiO$_2$ microparticles (5 mg ml$^{-1}$) on an ITO substrate (1 × 2 cm$^2$) and dried overnight. The Pt working electrode was prepared by sputtering a Pt layer (50 nm) on an ITO substrate (1 × 2 cm$^2$). Tafel measurements were recorded at a scan rate of 5 mV s$^{-1}$ under UV-light irradiation on the working electrodes (1 × 1 cm$^2$ immersed area) in ultra-pure water using a Metrohm AUTOLAB potentiostat.

Transient photocurrent response measurements were performed in the three-electrode configuration with samples as the working electrode, a Pt wire as the counter electrode, and an Ag/AgCl electrode (1 M KCl) as the reference electrode. The working electrodes were prepared by dropping 100 μl of an aqueous suspension of MXene-derived γ-Fe$_2$O$_3$/Pt/TiO$_2$ microrobots before and after nanoplastics' capture on an ITO substrate (1 × 2 cm$^2$) and dried overnight (capture experiments conditions: 0.75 mg ml$^{-1}$ microrobots, ~10$^{12}$ nanoplastics ml$^{-1}$, UV-light irradiation for 5 min, microrobots' collection using a neodymium magnet). The photocurrent was measured at 0 V vs. OCP by turning on and off the UV-light irradiation at time intervals of 10 s.

**Numerical simulation**. The numerical simulation was performed using the transport of diluted species module of the COMSOL Multiphysics software. For the simulation, the cross-section of the MXene-derived Pt/TiO$_2$ microrobots was considered. To emulate the MXene structure, the TiO$_2$ microparticles were designed as 10 rectangles with a size of 100 nm × 2 μm and inter-distance of 100 nm, placed inside an 8 × 4 μm$^2$ rectangle. Pt deposition can occur on the multi-layered side or the flat side of the TiO$_2$ microparticles. Pt deposition on the multi-layered side was designed using 100 × 50 nm$^2$ rectangles attached to the TiO$_2$ rectangles on their left side. Pt deposition on the flat side was designed using a 50 nm × 2 μm rectangle attached to the last TiO$_2$ rectangle on the right. The H$_2$ generation at the interface between Pt and water due to the photogenerated electron-hole pairs in TiO$_2$ was simulated by choosing water/Pt boundaries as the generation ones. To calculate H$_2$ diffusion for 0.1 s light irradiation, an H$_2$ diffusion coefficient in water at 25 °C of 5.11 × 10$^{-9}$ m$^2$ s$^{-1}$ was used, while the photo-generation rate was set to 1 mmol m$^{-2}$ s$^{-1}$.

**Nanoparticle tracking analysis**. Carboxylated polystyrene nanoparticles (50 nm) were used as a model for nanoplastics in water. Nanoplastics' concentration measurements were performed using the Malvern Panalytical NanoSight LM10 nanoparticle tracking system. The instrument was configured with a 532 nm laser diode and sCMOS camera and 20× optical objective. The nanoplastics' suspension was diluted with ultra-pure water and injected into the NanoSight cell for analysis. NTA 3.4 software was used to capture videos of nanoplastics at a constant temperature of 25 °C and a frame rate of 30 fps. After tracking the recorded videos under the same parameters, the software returned nanoplastics' size distribution through finite track length adjustment (FTLA) and concentration per ml.

**Nanoplastics' capture experiments**. For nanoplastics' quantification by NTA, a stock solution of nanoplastics was prepared by diluting the commercial suspension by a factor of 50000 using pH 3 water. In a typical experiment, the microrobots (0.75 mg) were added to 1 ml of nanoplastics' suspension in a UV-transparent cuvette placed inside a customized irradiation chamber and exposed to the UV-light emitted by three LZ4-04UV00 365 nm UV LEDs for different durations (1, 3, and 5 min). At the end of the treatment, microrobots were magnetically separated from the solution using a neodymium magnet. Both microrobots and the treated solutions were stored for further analyses. Each experiment has been repeated at least three times. The same experiments were performed using γ-Fe$_2$O$_3$/Ti$_3$C$_2$T$_x$ MXene microparticles as a reference or the microrobots without UV-light irradiation. Additional experiments have been conducted at different nanoplastics' concentrations (~10$^6$, 10$^{12}$, 10$^{14}$ nanoplastics ml$^{-1}$).

The removal capacity $q_t$ [mg g$^{-1}$] was calculated according to the following equation

$$q_t = \frac{C_0 - C_t}{m} V \qquad (1)$$

where $C_0$ and C$_t$ [mg ml$^{-1}$] are the nanoplastics' concentration at 0 min and time $t$, respectively, $V$ [ml] is the volume of the solution (1 ml), and $m$ [g] is the sample's mass (7.5 × 10$^{-4}$ g). Nanoplastics' concentration values measured by NTA [ml$^{-1}$] were converted to nanoplastics' concentration values $C_0$ and $C_t$ [mg ml$^{-1}$] using the estimated mass for the single nanoplastic (Supplementary Discussion 1).

For the reusability test, after each capture experiment, the microrobots were vigorously agitated in water at pH 11 for 10 min to induce the release of the captured nanoplastics.

**Electrochemical impedance spectroscopy measurements**. EIS measurements were performed at an applied potential of 0 V vs. ref. with a 10 mV superimposed sinusoidal root-mean-square voltage in the frequency range 10$^5$ ÷ 10$^0$ Hz in 100 μl electrolyte solution consisting of ultra-pure water with 10 mM K$_3$Fe(CN)$_6$ as the redox probe, using SPEs connected to a Metrohm AUTOLAB potentiostat. Before EIS measurements with microrobots, a 100 μl aqueous suspension of microrobots (or microrobots with captured nanoplastics) was spotted drop-by-drop on the working electrode area of the SPE. A neodymium magnet placed on the SPE backside accelerated microrobots' landing and allowed them to stay onto the SPE while replacing water with the electrolyte. The EIS experiment with nanoplastics was carried out using the MXene-derived γ-Fe$_2$O$_3$/Pt/TiO$_2$ microrobots collected with a neodymium magnet after the following capture experiments: 0.75 mg ml$^{-1}$ microrobots, ~10$^6$, 10$^{12}$, and 10$^{14}$ nanoplastics ml$^{-1}$, UV-light irradiation for 1 min. There was no exposure to UV-light during the EIS measurements.

## Data availability

The data generated in this study are provided in the manuscript or its Supplementary Information or Figshare repository (https://doi.org/10.6084/m9.figshare.19904512).

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

## Acknowledgements

M.P. was supported by Ministry of Education, Youth and Sports (Czech Republic) grant LL2002 under ERC CZ program. M.Ur. acknowledges the financial support by the European Union's Horizon 2020 research and innovation program under the Marie Skłodowska-Curie grant agreement No. 101038066. This work was supported by the project Advanced Functional Nanorobots (reg. No. CZ.02.1.01/0.0/0.0/15_003/0000444 financed by the EFRR). This work was supported by the ESF under the project CZ.02.2.69/0.0/0.0/18_053/0016962. CzechNanoLab project LM2018110 funded by MEYS CR is gratefully acknowledged for the financial support of the measurements/sample fabrication at CEITEC Nano Research Infrastructure.

## Author contributions

M.Ur. designed, prepared, and characterized the microrobots, performed the analysis of the motion, the numerical simulation, and the electrochemical measurements, and wrote the manuscript. M.Us. synthesized γ-Fe₂O₃ nanoparticles and conducted nanoplastics' capture experiments. M.Ur. and M. Us. carried out NTA with F.N.'s participation. M.Ur. and M.P. originated the idea. M.P. supervised the research. All authors have given approval to the final version of the manuscript.

## Competing interests

The authors declare no competing interests.
