## [Peer Review File · Nature Communications]

REVIEWER COMMENTS

Reviewer #1 (Remarks to the Author):

In this article, the authors reported multifunctional MXene-based microrobots for “on-the-fly” capture and detection of nanoplastics. They fabricated MXene-based Fe₃O₄/Pt/TiO₂ microrobots through a thermal annealing process of exfoliated Ti₃C₂T_x MXene microparticles. The micromotors can be propelled by UV-light with quick on/off switching. It is interesting that the MXene-based Fe₃O₄/Pt/TiO₂ microrobots can move in 2D and in 3D, which is related the way of Pt coating. In addition, the ability of micromotors to trap and detect nanoplastics was evaluated. Thus, the manuscript can be published after the addressing the following questions.

1. In this article, why the Pt coating was applied to fabricate Fe₃O₄/Pt/TiO₂ microrobots? What was the role of Pt coating, just to create the asymmetry of the micromotor?
2. The authors should explain how the Fe₃O₄ particles are modified to the micromotors in the text and scheme.
3. In the explanation of difference in speed between the micromotors moving in 2D and in 3D, the authors claimed that there is a probability of 67% to have the layered side of the MXene-based TiO₂ microparticles as the one exposed to the Pt target, and a 33% to have the layered side exposed. How do you get 67% and 33%?
4. The authors should explain in more detail why different position of Pt coating lead to the different motion behavior (moving in 2D and 3D).
5. For the nanoplastics capture using Fe₃O₄/Pt/TiO₂ microrobots, the capture experiment of nanoplastics by microrobots with or without UV light should be tested.
6. All the nanoplastics capture experiments have been conducted in pH 3 water. So, the motion behavior of the Fe₃O₄/Pt/TiO₂ microrobots should be given in the corresponding section.
7. For the nanoplastics detection, the authors should measure the impedance change of the micromotors after adsorption different concentrations of nanoplastics to verify the reliability of the method. And, the effect of micromotors' movement on impedance should also be measured.
8. The authors should check the manuscript carefully. The scale bar in the videos is missing. The format of references should be corrected.

Reviewer #2 (Remarks to the Author):

I have carefully read this paper entitled “Trapping and Detecting Nanoplastics by MXene-based Microrobots”. This work designed an efficient strategy for the removal and detection of the nanoplastics in water based on innovative self-propelled light-powered magnetic MXene-based microrobots. The MXene-based Fe₃O₄/Pt/TiO₂ microrobots exhibit strong mobility in fuel-free water and have great potential for the removal and degradation of other organic pollutants. This work can be reconsidered with major revisions. Specific comments are given below:

1. XRD patterns are necessary to prove the successful fabrication of MXene-based Fe₃O₄/Pt/TiO₂.
2. In addition to SEM and EDX, more characterizations are needed to determine the morphological and microstructural information of the Pt layer and Fe₃O₄ nanoparticles in Fe₃O₄/Pt/TiO₂.
3. Please explain that a continuous Pt coating produces stronger micro/nanorobots propulsion.
4. Where does the powerful driving force of the microrobots come from?
5. Curiously, there is no diffraction peak of Ti₃C₂ in the XRD pattern.
6. Additional proof should be provided for the phenomenon that the Rct significantly rises after loading the microrobots onto the SPE compared to the microrobots with nanoplastics.
7. Several measurements, such as photoluminescence spectra and transient photocurrent response, can be performed to study the charge separation and transfer behaviors of the microrobot after capturing nanoplastics.

Reviewer #3 (Remarks to the Author):

The authors report on microrobots of Fe₃O₄/Pt/TiO₂ for trapping and detecting nanoplastics from water. Thanks to the semiconductor TiO₂ and Pt coating, the resulted microrobots showed 6 degrees of freedom under UV light. The microrobots are positively charged and the nanoparticle of polymers are negatively charged and therefore they are attracted to each other, then the Fe₃O₄ can be utilized to collect the robots by magnetism for further detection studies. The authors used a number of characterization techniques (e.g., XRD, SEM, TEM, XPS, NTA, EIS) to study the synthesized microrobots and their behavior. The study is interesting, but it suffers from many weaknesses such as:

- 1- The use of MXene in this study is not justified at all considering the target is TiO₂ that is then get coated by Pt. It is a very expensive route to make TiO₂, while layered titanate can be made from much cheaper precursors than MXenes. It is not clear what is the value of using MXene in this study at all. A control sample using layered titanate should be used.

2- The use “MXene-based” should be removed from everywhere in the manuscript since based on XRD there is no MXene in these microrobots and they are mainly TiO₂. If the authors would like to keep MXene somewhere in the title, it should be “MXene-Derived Oxide”.

3- The justification for using multilayer MXene is not clear, the whole mechanism depends on surface charge, and multilayer MXene has a very low specific surface area and therefore the resulted oxide would have a low specific surface area too. The specific surface area of MXene and the derived microrobot should be measured

4- The claim of nanoparticles trapped between layers should be revised considering. The nanoparticles can only access the external surface of the oxidized MXene and the macropores slits between the formerly multilayer stacks of MXenes. The volume of macropores is significantly reduced after oxidation as shown in figure S1b compared to S1a most. Also, the use of “layers” here is confusing since it relates to the 2D layers of MXenes where the interlayer spacing is <2 nm and can't be accessed by the 50 nm plastic particles.

5- The lack of performance/behavior comparisons with other microrobots prepared differently make it very difficult for the reader to appreciate the advancement of this work. More comparisons with the literature is necessary for the discussion. Is it possible to re-use the microrobots?

6- Any claim about removal capability should be supported by capacity of removal mg/g, the 97% efficiency is meaningless without such quantification. Considering that surface area of multilayer MXene is only in the range of 10 m²/g, it will be surprising if the resulted composite has high removal capacity since it depends on surface charge. If the removal capacity can't be given, then such claims should be removed and the manuscript should focus on detection.

7- The EIS data in fig 6a is not presented correctly. Nyquist plot shape is meaningless unless the graph is square with equivalent axes.

8- While one can assume that all of the results in the paper relate to the 0 min thermal annealing condition, since the authors note that it preserved the layered structure best, it is not clear that all subsequent figures/results relate to the 0 min annealing condition.

Response to the Reviewer #1

In this article, the authors reported multifunctional MXene-based microrobots for “on-the-fly” capture and detection of nanoplastics. They fabricated MXene-based Fe₃O₄/Pt/TiO₂ microrobots through a thermal annealing process of exfoliated Ti₃C₂T_x MXene microparticles. The micromotors can be propelled by UV-light with quick on/off switching. It is interesting that the MXene-based Fe₃O₄/Pt/TiO₂ microrobots can move in 2D and in 3D, which is related the way of Pt coating. In addition, the ability of micromotors to trap and detect nanoplastics was evaluated. Thus, the manuscript can be published after the addressing the following questions.

1. In this article, why the Pt coating was applied to fabricate Fe₃O₄/Pt/TiO₂ microrobots? What was the role of Pt coating, just to create the asymmetry of the micromotor?

Author reply: The role of the Pt coating was to break the symmetry of the MXene-derived TiO₂ microparticles and improve the charge separation at the Pt/TiO₂ interface. Then, it resulted in microrobots' motion under UV-light irradiation in fuel-free water *via* the self-electrophoretic mechanism, whose description in the manuscript has been improved according to question n. 4 of Reviewer #2. It must be noted that the MXene-derived TiO₂ microparticles have a sort of intrinsic asymmetry due to their multi-layered structure, which was sufficient to induce their movement under UV-light irradiation in the presence of relatively high concentrations of H₂O₂ (>1%). This behavior is not surprising since self-propelled single-component microrobots have already been reported. However, this motion was not particularly attractive because of the low efficiency (only a minor fraction of particles could move), low speed, and required H₂O₂. For these reasons, we preferred to apply the Pt coating, which led to higher efficiency, speed, fuel-free self-propulsion, and 3D motion behavior. The Reviewer's comment stimulated us to mention this unsatisfactory experiment (i.e., microrobots motion without Pt) on page 10 of the manuscript as follows.

“It was noticed that, without the Pt layer, some MXene-derived TiO₂ microparticles could move under UV-light irradiation in the presence of relatively high H₂O₂ concentrations (>1% H₂O₂) due to their asymmetric structure. However, their low speed and the required toxic H₂O₂ made them less attractive than the MXene-derived γ -Fe₂O₃/Pt/TiO₂ microrobots, whose motion did not require fuel or surfactant.”

2. The authors should explain how the Fe₃O₄ particles are modified to the micromotors in the text and scheme.

Author reply: It is worth noting that, thanks to the comments n. 1 and 2 from Reviewer #2, we realized that the as-synthesized magnetic nanoparticles mainly consisted of γ -Fe₂O₃. Then, as requested by the Reviewer, the following sentence, explaining how microrobots' surface was modified with the γ -Fe₂O₃ nanoparticles, has been added on page 6 of the manuscript.

“MXene-derived TiO₂ microparticles were asymmetrically covered with a 50 nm thick Pt layer by sputtering and mixed with γ -Fe₂O₃ nanoparticles for 1 h at room temperature, fabricating light-powered magnetic microrobots.”

Fig. 1(a), reporting the scheme of microrobots' fabrication steps, has been modified as follows.

Figure 1. Fabrication and characterization of the MXene-derived $\gamma\text{-Fe}_2\text{O}_3/\text{Pt}/\text{TiO}_2$ microrobots. (a) Scheme of the fabrication steps.

3. In the explanation of difference in speed between the micromotors moving in 2D and in 3D, the authors claimed that there is a probability of 67% to have the layered side of the MXene-based TiO_2 microparticles as the one exposed to the Pt target, and a 33% to have the layered side exposed. How do you get 67% and 33%?

Author reply: We ideally assumed the MXene-derived TiO_2 microparticle as a cube/parallelepiped formed by several adjacent layers. Therefore, at a first approximation, 4 out of 6 cube/parallelepiped's faces appear discontinuous due to its layered structure, while the other 2 appear flat. Pt sputtering deposition is conducted on substrates placed in front of the Pt target inside the sputtering machine. Specifically, a microparticles suspension (5 mg ml^{-1}) was dropped on the glass slides and dried overnight to prepare the substrates. During the dropping step, we assumed that the particles randomly deposit on the glass slides, having a probability of $\sim 67\%$ ($4/6$) or $\sim 33\%$ ($2/6$) to expose one of the multi-layered or flat faces to the Pt target, respectively. This concept has been better explained on page 13 of the manuscript as follows.

“By ideally assuming these microparticles as cubes/parallelepipeds formed by several adjacent layers, it is observed that four faces out of six are layered and discontinuous. At the same time, the remaining two are flat and continuous. Pt deposition by sputtering technique is conducted on substrates placed in front of the Pt target. These substrates consist of microscope glass slides on which a suspension of MXene-derived TiO_2 microparticles is dropped and dried overnight to form a monolayer of particles. During this preliminary preparation step, it can be assumed that the particles randomly deposit on the glass slides. So, in the ideal case, there is a probability of $\sim 67\%$ and $\sim 33\%$ to have one of the multi-layered or flat faces like the one exposed to the Pt target, as depicted in Fig. 3(c).”

4. The authors should explain in more detail why different position of Pt coating lead to the different motion behavior (moving in 2D and 3D).

Author reply: We believe that the 3D motion originates from a negative photogravitactic behavior, i.e., a microrobots' movement against gravity due to vertical light-irradiation (in our experiments, from the bottom of the glass slide) (references [49-51]). To observe this phenomenon, the microrobots' speed and, consequently, the driving force should be potent enough to overcome, in combination with the buoyancy force, the gravitational force exerted on the microrobots, allowing the movement of the microrobots also in the upward direction. Therefore, we believe that the difference in the 2D and 3D motion may be traced back to their different speeds and, ultimately, to the different Pt coatings. Specifically, we believe that those moving on the focal plane (2D motion) are the microrobots with the discontinuous Pt coating, whose speed and driving force are lower. Indeed, the continuous Pt coating obtained on the flat faces of the MXene-derived microparticles produces stronger propulsion than the discontinuous Pt coating on the multi-layered faces. This assumption is supported by our numerical simulation as well as relevant published works (references [27], [53], and [54] in the manuscript). To better explain the origin of the microrobots' movement at the basis of the 3D motion, the discussion on page 10 of the manuscript was improved as follows.

“This phenomenon is known as negative photogravitaxis, i.e., a particles’ migration against gravity when irradiated vertically from the bottom of the substrate, which has been previously observed for other photochemical micromotors.^{49–51}”

References [49-51] have been included in the references list.

- [49] Zhou, C., Zhang, H. P., Tang, J. & Wang, W. Photochemically Powered AgCl Janus Micromotors as a Model System to Understand Ionic Self-Diffusiophoresis. *Langmuir* **34**, 3289–3295 (2018).
- [50] Singh, D. P., Uspal, W. E., Popescu, M. N., Wilson, L. G. & Fischer, P. Photogravitactic Microswimmers. *Adv. Funct. Mater.* **28**, 1706660 (2018).
- [51] Zhang, J. *et al.* Photochemical micromotor of eccentric core in isotropic hollow shell exhibiting multimodal motion behavior. *Appl. Mater. Today* **26**, 101371 (2022).

Furthermore, the photogravitaxis has been mentioned in the abstract, introduction, and conclusions sections.

The discussion about the powerful thrusting force of the microrobots with a continuous Pt coating has been improved (see also the reply to question n. 3 of Reviewer #2). Additionally, we expanded the discussion about 3D motion ability on page 14 of the manuscript by adding the following paragraph.

“...a continuous Pt deposition on the flat side of the MXene-derived TiO₂ microparticle may be the origin of the higher speed and stronger propulsive force of the resulting microrobots. This powerful driving force, combined with the buoyancy, overcomes the gravitational force and unlocks the microrobots’ movement in the upward direction, leading to their 3D motion ability.”

5. For the nanoplastics capture using Fe₃O₄/Pt/TiO₂ microrobots, the capture experiment of nanoplastics by microrobots with or without UV light should be tested.

Author reply: As requested by the Reviewer, we performed the nanoplastics’ capture experiment using the microrobots without UV-light irradiation. Supplementary Fig. 10 compares the obtained results with the MXene and microrobots under UV-light irradiation. Microrobots’ performance in the absence of UV-light irradiation (i.e., in static condition) is close to that of the MXene, evidencing the critical contribution of microrobots’ self-propulsion to the nanoplastics’ capture process.

Supplementary Figure 2. Nanoplastics' concentration in the treated suspensions as a function of the treatment's duration (0, 1, 3, and 5 min) using $\gamma\text{-Fe}_2\text{O}_3/\text{Ti}_3\text{C}_2\text{T}_x$ MXene microparticles under UV-light irradiation, and MXene-derived $\gamma\text{-Fe}_2\text{O}_3/\text{Pt}/\text{TiO}_2$ microrobots (thermal annealing condition: 0 min at 550°C) with and without UV-light irradiation (other capture experiments conditions: 0.75 mg ml⁻¹ samples, 6x10⁹ nanoplastics ml⁻¹, water at pH 3, samples' collection using a neodymium magnet). Error bars represent the standard deviation, n = 3 independent replicates.

Supplementary Fig. 10 has been added to the Supplementary information and commented on page 10 of the manuscript as follows.

“Control experiments using the microrobots without UV-light irradiation, i.e., in the static condition, revealed a trend similar to the MXene microparticles (Supplementary Fig. 10), further confirming the crucial contribution of microrobots' self-propulsion to the nanoplastics capture process.

6.All the nanoplastics capture experiments have been conducted in pH 3 water. So, the motion behavior of the Fe3O4/Pt/TiO2 microrobots should be given in the corresponding section.

Author reply: We agree with the Reviewer's comment. The motion behavior of the microrobots in water at pH 3 has been studied, and the measured speed values are reported in Supplementary Fig. 9. Once more, microrobots moving on the *xy* plane (2D motion) and in *xyz* space (3D motion) were observed. The average speeds for the 2D and 3D motion at pH 3 decreased significantly compared to pH 7 (Fig. 3(a) in the manuscript). This is attributed to the higher ionic strength of water at pH 3 or a lower photochemical activity (reference [58] in the manuscript). Nevertheless, 13% of the recorded microrobots could undergo the 3D motion at pH 3. We also noted a higher tendency to stick to the surface of the glass slide due to the lower repulsion between the microrobots and the glass slide at acidic pH (reference [58] in the manuscript).

Supplementary Figure 3. Speed distributions of the MXene-derived $\gamma\text{-Fe}_2\text{O}_3/\text{Pt}/\text{TiO}_2$ microrobots (thermal annealing condition: 0 min at 550°C) showing 2D and 3D motion in fuel-free water at pH 3 under UV-light irradiation.

Supplementary Fig. 9 has been added to the Supplementary information and commented on page 17 of the manuscript as follows.

“Before, microrobots’ motility in water at pH 3 was studied. About 13% of the recorded microrobots show the 3D motion, although at a lower average speed than pH 7 (Supplementary Fig. 9). Also, the speed of the 2D motion decreases significantly. This behavior is explained by the higher ionic strength of water at pH 3 or the lower microrobots’ photochemical activity.⁵⁸ In addition, the attraction between the positively charged microrobots and the negatively charged glass slide may contribute to slowing down the microrobots, which more often are stuck to the substrate.”

7. For the nanoplastics detection, the authors should measure the impedance change of the micromotors after adsorption different concentrations of nanoplastics to verify the reliability of the method. And, the effect of micromotors’ movement on impedance should also be measured.

Author reply: Following the Reviewer’s comment, the microrobots’ impedance after adsorbing different nanoplastics’ concentrations has been measured (capture experiment conditions: 0.75 mg ml^{-1} microrobots, $\sim 10^6$, 10^{12} , and 10^{14} nanoplastics ml^{-1} , UV-light irradiation for 1 min, microrobots’ collection using a neodymium magnet). The measured Nyquist plot and Bode plots are reported below in Supplementary Fig. 16. Impedance data were fitted using the equivalent circuit model illustrated in Fig. 6(b) in the manuscript. It was found that the R_{ct} for the bare microrobots and microrobots exposed to $\sim 10^6$ nanoplastics ml^{-1} were almost the same ($\sim 5000 \Omega$). Indeed, such nanoplastics concentration is too low to produce a detectable impedance variation. Upon exposure to $\sim 10^{12}$ nanoplastics ml^{-1} , the R_{ct} increases significantly up to $\sim 9000 \Omega$, as shown in the first version of the manuscript. However, a further

rise in R_{ct} ($\sim 18000 \Omega$) was noted after treating a solution with $\sim 10^{14}$ nanoplastics ml^{-1} . These experiments indicate that the proposed methodology is reliable for concentrations higher than $\sim 10^6$ nanoplastics ml^{-1} . We could not estimate an upper limit as $\sim 10^{14}$ nanoplastics ml^{-1} constitutes the concentration of our polystyrene nanoparticles' stock solution. It is worth noting that a careful investigation of the other capture experiment parameters, such as the microrobots' concentration and treatment duration, may reduce the sensor's detection limit. We will consider these aspects in future works.

Supplementary Figure 4. Detection of nanoplastics at different concentrations by electrochemical impedance spectroscopy (EIS) technique after the preconcentration with the MXene-derived $\gamma\text{-Fe}_2\text{O}_3/\text{Pt}/\text{TiO}_2$ microrobots. (a) Nyquist plots showing the impedance real ($\text{Re}(Z)$) and imaginary ($-\text{Im}(Z)$) parts as a function of the frequency and (b) Bode plots showing the impedance modulus ($|Z|$) and phase ($-\text{Phase}(Z)$) as a function of the frequency for a screen-printed electrode (SPE) after MXene-derived $\gamma\text{-Fe}_2\text{O}_3/\text{Pt}/\text{TiO}_2$ microrobots' (thermal annealing condition: 0 min at 550°C) loading, and SPEs after loading microrobots with captured nanoplastics at different concentrations (capture experiment conditions: 0.75 mg ml^{-1} microrobots, $\sim 10^6$, 10^{12} , and 10^{14} nanoplastics ml^{-1} , water at pH 3, UV-light irradiation for 1 min, microrobots collection using a neodymium magnet). The EIS measurements were performed in a $10 \text{ mM Fe(CN)}_6^{4-/3-}$ aqueous solution at 0 V vs. ref. with a 10 mV superimposed sinusoidal root-mean-square voltage in the frequency range 10^5 - 10^0 Hz. The lines represent fits to the data based on the equivalent circuit model reported in Fig. 6(b) in the manuscript. (c) R_{ct} values obtained from the Nyquist plots fit. Error bars represent the fitting error.

The following paragraph has been added to the discussion on page 21 of the manuscript. The methods section has been updated accordingly. Supplementary Fig. 16 has been included in the Supplementary information. Supplementary Table 2, showing the EIS fitting results, has been updated.

“The method's reliability has been verified by measuring the impedance of microrobots exposed to different nanoplastics concentrations ($\sim 10^6$ and 10^{14} nanoplastics ml^{-1}). Supplementary Fig. 16 shows the recorded Nyquist and Bode plots, whose fitting parameters are reported in Supplementary Table 2. The R_{ct} for the bare microrobots is similar to microrobots with 10^6 nanoplastics ml^{-1} , suggesting that this low concentration can not produce a detectable impedance variation. On the other hand, a remarkable increase of the R_{ct} ($\sim 18000 \Omega$) was obtained for the microrobots with 10^{14} nanoplastics ml^{-1} , corroborating the consistency of the proposed sensing strategy.

On these bases, the microrobots quickly accumulate nanoplastics in water and allow determining their presence at concentrations higher than 10^6 nanoplastics ml^{-1} through impedance measurements.”

Regarding the effect of microrobots’ movement on the impedance, it must be noted that the EIS measurements were performed only after the capture experiments. In particular, first, we captured nanoplastics using the self-propelled microrobots under UV-light irradiation for 1 min. Then, the microrobots were collected using a neodymium magnet and transferred to the SPEs to measure the impedance. There was no exposure to UV-light and, thus, microrobots’ movement during EIS measurement. Actually, microrobots were firmly adhering to the SPE due to the presence of a neodymium magnet on the backside of the electrode, as illustrated in Fig. 6(d) in the manuscript. Nevertheless, we thank the Reviewer for this comment, which suggested that this protocol was not clear enough in the previous version of the manuscript. Therefore, we added the following sentence to the methods section on page 28 of the manuscript.

“There was no exposure to UV-light during the EIS measurements.”

8.The authors should check the manuscript carefully. The scale bar in the videos is missing. The format of references should be corrected.

Author reply: We thank the Reviewer. As requested, the scale bar has been added to the videos. Moreover, we noted that the scale bar was also missing in the frames of NTA videos reported in the previous version of Fig. 4(c) and Fig. 5(d). Finally, the references’ format has been corrected.

Response to the Reviewer #2

I have carefully read this paper entitled “Trapping and Detecting Nanoplastics by MXene-based Microrobots”. This work designed an efficient strategy for the removal and detection of the nanoplastics in water based on innovative self-propelled light-powered magnetic MXene-based microrobots. The MXene-based Fe₃O₄/Pt/TiO₂ microrobots exhibit strong mobility in fuel-free water and have great potential for the removal and degradation of other organic pollutants. This work can be reconsidered with major revisions. Specific comments are given below:

1. XRD patterns are necessary to prove the successful fabrication of MXene-based Fe₃O₄/Pt/TiO₂.

Author reply: We agree with the Reviewer’s comment. The XRD pattern of the MXene-derived TiO₂ microparticles after Pt deposition and magnetic nanoparticles loading was measured and reported in Supplementary Fig. 6. The characteristic peaks of Pt(111) and (220) are visualized in the final sample pattern. However, the diffraction peak at 35.2° does not allow to distinguish the magnetite (Fe₃O₄) crystalline phase from the maghemite (γ-Fe₂O₃), which was identified by XPS (see comment n. 2 from the same Reviewer). Still, this result proves the successful fabrication of MXene-derived γ-Fe₂O₃/Pt/TiO₂ microrobots, in agreement with SEM-EDX, XPS, and VSM analyses.

Supplementary Figure 5. XRD patterns of the MXene-derived TiO₂ microparticles and MXene-derived γ-Fe₂O₃/Pt/TiO₂ microrobots (thermal annealing condition: 0 min at 550°C).

Supplementary Fig. 6 has been included in the Supplementary information and mentioned on page 8 of the manuscript as follows.

“Besides, the XRD pattern of the final sample exhibited Pt and γ-Fe₂O₃ characteristics peaks (Supplementary Fig. 6).”

2. In addition to SEM and EDX, more characterizations are needed to determine the morphological and microstructural information of the Pt layer and Fe₃O₄ nanoparticles in Fe₃O₄/Pt/TiO₂.

Author reply: In addition to SEM and EDX images in Fig. 1 in the manuscript, we recorded an SEM image of a cluster of magnetic nanoparticles (Supplementary Fig. 3), whose size was <50 nm. Moreover, based on the previous comment, the microrobots’ XRD pattern was measured (Supplementary Fig. 6), finding the characteristic diffraction peaks of Pt. Regarding the magnetic nanoparticles, the diffraction peak at 35.2° does not allow to distinguish the magnetite

(Fe_3O_4) crystalline phase from the maghemite ($\gamma\text{-Fe}_2\text{O}_3$). To better characterize the Pt layer and, in particular, the magnetic nanoparticles, we investigated their chemical composition by measuring the microrobots' XPS spectrum. Supplementary Fig. 7(a) reports the XPS survey spectra, where the peak of Pt 4f and Fe 2p are observed. High-resolution XPS spectra of Pt 4f and Fe 2p have also been measured and shown in Supplementary Fig. 7(b) and (c), respectively. Pt 4f XPS spectrum fitting confirmed the deposition of Pt^0 (reference [3] in the Supplementary information). Fe 2p XPS spectrum fitting was only compatible with $\gamma\text{-Fe}_2\text{O}_3$ (reference [4] in the Supplementary information). Therefore, we wish to thank the Reviewer for suggesting to improve the materials' characterization, revealing that the phase of the synthesized magnetic nanoparticles is mainly maghemite rather than magnetite.

Supplementary Figure 6. SEM image of a cluster of $\gamma\text{-Fe}_2\text{O}_3$ nanoparticles. The scale bar is 500 nm.

Supplementary Figure 7. (a) XPS survey spectrum of the MXene-derived $\gamma\text{-Fe}_2\text{O}_3/\text{Pt}/\text{TiO}_2$ microrobots (thermal annealing condition: 0 min at 550°C). (b) Pt 4f and (c) Fe 2p high-resolution XPS spectra.

The chemical formula “ Fe_3O_4 ” has been replaced by “ $\gamma\text{-Fe}_2\text{O}_3$ ” throughout the manuscript, Supplementary information, and response letter. Supplementary Fig. 3 and 6 have been added to the Supplementary information and commented on pages 6 and 8 of the manuscript as follows. The binding energy values for all fitted peaks have been reported in Supplementary Table 1.

“The latter can not be directly visualized on the microrobots due to the high surface roughness of the MXene-derived TiO_2 microparticles and their small size (<50 nm), as indicated by the SEM image of a cluster of $\gamma\text{-Fe}_2\text{O}_3$ nanoparticles (Supplementary Fig. 3).”

“XPS analysis confirmed the presence of Pt^0 and $\gamma\text{-Fe}_2\text{O}_3$ on the microrobots' surface (Supplementary Fig. 7), further proving the successful fabrication of MXene-derived $\gamma\text{-Fe}_2\text{O}_3/\text{Pt}/\text{TiO}_2$ microrobots.”

References [3] and [4] have been included in the reference list in the Supplementary information.

- [3] Bera, P. *et al.* Ionic dispersion of Pt over CeO₂ by the combustion method: Structural investigation by XRD, TEM, XPS, and EXAFS. *Chem. Mater.* **15**, 2049–2060 (2003).
- [4] Biesinger, M. C. *et al.* Resolving surface chemical states in XPS analysis of first row transition metals, oxides and hydroxides: Cr, Mn, Fe, Co and Ni. *Appl. Surf. Sci.* **257**, 2717–2730 (2011).

3. Please explain that a continuous Pt coating produces stronger micro/nanorobots propulsion.

Author reply: Previous works have demonstrated that the Pt coating significantly influences the motion behavior of Pt/semiconductor microrobots under UV-light irradiation. Specifically, it has been shown that Pt deposition on smooth microparticles produced a continuous Pt coating, while a discontinuous Pt coating is observed on rough microparticles. Then, the microrobots with a continuous Pt layer displayed higher speed under UV-light irradiation in H₂O₂ (reference [53] in the manuscript) or even the fuel-free motion ability (reference [27] in the manuscript). Moreover, a recent study showed that the continuous Pt film on SiO₂ microspheres was transformed into discrete Pt nanoparticles upon thermal annealing, causing an ~80% decrease in the self-electrophoretic speed of the microrobots (reference [54] in the manuscript). As confirmed by numerical simulations, this result was attributed to the lower electric field or fluid flow generated by the discontinuous Pt coating. On these bases, we believe that the stronger propulsion observed for some of our microrobots, leading to the 3D motion, is related to the continuous Pt coating on the flat sides of the intrinsically asymmetric MXene-derived TiO₂ microparticles. As requested by the Reviewer, this concept has been better explained on pages 13 and 14 of the manuscript by adding the following paragraphs.

“In fact, for UV-light-powered Pt/ZnO Janus micromotors, it has been demonstrated that a smooth ZnO microparticle results in a continuous Pt coating and the fuel-free motion ability, while a rough ZnO microparticle leads to a discontinuous Pt coating which requires H₂O₂ fuel for the autonomous propulsion.²⁷ Similarly, smooth TiO₂ microparticles coated by a continuous Pt layer exhibited a higher speed under UV-light-irradiation in H₂O₂ than rough ones.⁵³ Moreover, it has been shown that transforming a continuous Pt film on SiO₂ microspheres into discrete Pt nanoparticles *via* thermal annealing causes an ~80% decrease in their self-electrophoretic speed due to the lower electric field or fluid flow generated by the discontinuous Pt coating.⁵⁴”

“The continuous Pt coating on the flat side of the MXene-derived TiO₂ microparticle produces more H₂ compared to the discontinuous Pt coating on the multi-layered side, reflecting a larger H⁺ concentration gradient and, so, a higher speed and 3D motion ability. This conclusion is consistent with the previously reported stronger propulsion for microrobots coated by a continuous Pt layer.^{27,53,54}”

References [52] and [53] have been included in the references list.

- [53] Oral, C. M., Ussia, M., Yavuz, D. K. & Pumera, M. Shape Engineering of TiO₂ Microrobots for “On-the-Fly” Optical Brake. **18**, 2106271 (2021).
- [54] Lyu, X. *et al.* Active, Yet Little Mobility: Asymmetric Decomposition of H₂O₂ Is Not Sufficient in Propelling Catalytic Micromotors. *J. Am. Chem. Soc.* **143**, 12154–12164 (2021).

4. Where does the powerful driving force of the microrobots come from?

Author reply: The driving force of the microrobots is based on the generation of an asymmetric distribution of protons (H^+) upon UV-light irradiation, which establishes a local electric field inducing the motion of the charged microrobot *via* self-electrophoresis with TiO_2 as the forward side. As implicitly suggested by the Reviewer’s comment, we improved the description of this motion mechanism on page 13 of the manuscript as follows. In addition, the scheme of microrobots’ motion mechanism in Fig. 3(b) has also been upgraded, illustrating the H^+ concentration gradient and microrobots’ propulsion force, as shown below.

“Upon exposure to UV-light, electrons are promoted to the TiO_2 conduction band, leaving holes in the valence band. The metal/semiconductor Schottky junction at the Pt/TiO_2 interfaces facilitates charge separation. Electrons transferred from the TiO_2 conduction band to Pt and holes left in TiO_2 decompose water according to the reactions reported in Fig. 3(b). In particular, the H^+ ions generated at the TiO_2 side are consumed at the Pt side to produce H_2 , establishing an H^+ concentration gradient and, thus, a local electric field. This, in turn, drives Pt/TiO_2 micromotors motion via a self-electrophoretic mechanism with TiO_2 as the forward side.⁵²”

Figure 3. (b) Scheme of the general motion mechanism of Pt/TiO_2 microrobots (E_0 is the vacuum level, E_F is the Fermi level, Φ_{Pt} is Pt work function, Φ_B is the Schottky barrier height, E_C , E_V , E_g , and χ_{TiO_2} are TiO_2 conduction and valence band levels, optical bandgap and electron affinity).

5. Curiously, there is no diffraction peak of Ti_3C_2 in the XRD pattern.

Author reply: The Reviewer is right. Indeed, a previous work demonstrated that the same thermal annealing process ($550^\circ C$ in air) resulted in a TiO_2/Ti_3C_2 MXene core-shell structure, as confirmed by the presence of Ti_3C_2 peaks in the XRD pattern of the annealed sample (reference [43] in the manuscript). In our work, we varied the annealing conditions, including the time (0 min, 30 min, 60 min, and 120 min at $550^\circ C$ in air) and the temperature (0 min in 350, 450, and $550^\circ C$ in air, not shown in the manuscript). Unfortunately, no Ti_3C_2 peaks were observed in the diffraction patterns of the annealed samples. On this basis, residual Ti_3C_2 MXene was not claimed and further investigated. Accordingly, we defined our optimal sample (0 min at $550^\circ C$ in air) as an MXene-derived TiO_2 rather than a TiO_2/Ti_3C_2 composite. To clarify this point, the discussion of XRD patterns on page 6 of the manuscript was improved as follows.

“It is worth noting that Low and coworkers found small peaks attributed to Ti_3C_2 in the XRD pattern of the annealed sample and demonstrated a TiO_2/Ti_3C_2 core-shell structure by high-resolution TEM.⁴³ In this work, those XRD peaks were not found in the XRD pattern of any annealed sample, and thus the residual presence of the $Ti_3C_2T_x$ MXene after the thermal annealing process was not claimed and further investigated. Hence, the optimal sample was referred to as an MXene-derived TiO_2 rather than a TiO_2/Ti_3C_2 MXene composite.”

6. Additional proof should be provided for the phenomenon that the R_{ct} significantly rises after loading the microrobots onto the SPE compared to the microrobots with nanoplastics.

Author reply: Microrobots' loading on the SPE causes the R_{ct} increase due to the low conductivity of TiO_2 (reference [66] in the manuscript), which represents the main microrobots' component. The same behavior has also been observed after TiO_2 loading on FTO electrodes (reference [67] in the manuscript). Similarly, the R_{ct} further rises after loading microrobots with captured nanoplastics on the SPE due to the low conductivity of polystyrene (reference [68] in the manuscript). In addition, the negatively charged nanoplastics compensate the positively charged microrobots. Therefore, the attraction of the redox probe ions ($Fe(CN)_6^{4-/3-}$) is hindered, increasing the R_{ct} . To prove this explanation, the Zeta potential of microrobots with captured nanoplastics in water at pH 3 was measured and shown in Supplementary Fig. 15. Indeed, it was found that after nanoplastics' capture, the microrobots' Zeta potential decreased from +43 mV to -1 mV.

Supplementary Figure 8. Zeta potential of the MXene-derived $\gamma-Fe_2O_3/Pt/TiO_2$ microrobots (thermal annealing condition: 0 min at 550°C) after nanoplastics' capture in water at pH 3 (capture experiment conditions: 0.75 mg ml⁻¹ microrobots, $\sim 10^{12}$ nanoplastics ml⁻¹, water at pH 3, UV-light irradiation for 1 min, microrobots collection using a neodymium magnet).

Supplementary Fig. 15 was added to the Supplementary information and commented on page 21 of the manuscript as follows.

“The presence of the microrobots leads to a notable increase of the SPE's R_{ct} due to the low conductivity of the semiconducting TiO_2 , which represents the main constituent of the microrobots.⁶⁶ This behavior has also been observed after TiO_2 loading on FTO electrodes.⁶⁷ The R_{ct} further rises due to the lower microrobots' conductance in consequence of the inclusion of a large quantity of non-conducting polystyrene nanoparticles.⁶⁸ Additionally, nanoplastics' capture results in the neutralization of microrobots' positive surface charge and, thus, the Coulomb repulsion of the redox probe ions ($Fe(CN)_6^{4-/3-}$), as demonstrated by the microrobots' Zeta potential decrease (+43 \rightarrow -1 eV) after nanoplastics' capture (Supplementary Fig. 15). This mechanism, similar to the one at the base of DNA hybridization sensors, contributed to the R_{ct} increase.^{62,}”

References [66-68] have been comprised in the reference list.

[66] Yildiz, A., Lisesivdin, S. B., Kasap, M. & Mardare, D. Electrical properties of TiO_2 thin films. *J. Non. Cryst. Solids* **354**, 4944–4947 (2008).

[67] Luo, D., Liu, B., Gao, R., Su, L. & Su, Y. TiO₂/CuInS₂-sensitized structure for sensitive photoelectrochemical immunoassay of cortisol in saliva. *J. Solid State Electrochem.* **26**, 749–759 (2022).

[68] Qi, X. Y. *et al.* Enhanced electrical conductivity in polystyrene nanocomposites at ultra-low graphene content. *ACS Appl. Mater. Interfaces* **3**, 3130–3133 (2011).

7. Several measurements, such as photoluminescence spectra and transient photocurrent response, can be performed to study the charge separation and transfer behaviors of the microrobot after capturing nanoplastics.

Author reply: We agree with the Reviewer's comment. The photoluminescence of the microrobots before and after capturing nanoplastics has been measured using a 355 nm laser for the excitation. The recorded spectra are shown in Fig. R1. In the microrobots' spectrum, the peak at ~450 nm is attributed to the TiO₂ photoluminescence and agrees with previous works (reference [R1] in the response letter). After nanoplastics' capture, the photoluminescence spectrum exhibits a broad peak at ~550 nm, ascribed to the polystyrene nanoparticles (reference [R2] in the response letter). This result further confirmed the successful nanoplastics' capture by microrobots. However, after several tentatives of optimization, the measurements were still affected by high noise. Additionally, the signals above 650 nm are artifacts due to the utilized detector. Considering the limitation of our experimental apparatus, we prefer not to report and describe this experiment in the manuscript.

Figure R1. Photoluminescence spectra of the MXene-derived γ -Fe₂O₃/Pt/TiO₂ microrobots before and after capturing nanoplastics (capture experiment conditions: 0.75 mg ml⁻¹ microrobots, $\sim 10^{12}$ nanoplastics ml⁻¹, UV-light irradiation for 5 min, microrobots collection using a neodymium magnet). Samples were excited using a 355 nm laser.

[R1] Parnicka, P. *et al.* Influence of the preparation method on the photocatalytic activity of Nd-modified TiO₂. *Beilstein J. Nanotechnol.* **9**, 447–459 (2018).

[R2] Kim, E. *et al.* White light emission from polystyrene under pulsed ultra violet laser irradiation. *Sci. Rep.* **3**, 3–6 (2013).

The transient photocurrent response of the microrobots before and after capturing nanoplastics has been measured and shown in Supplementary Fig. 14. In both cases, rapid, stable, and repeatable photocurrent responses are observed by turning on and off the UV-light

at time intervals of 10 s. However, the photocurrent decreases by a factor of ~ 3 after nanoplastics' capture ($9 \rightarrow 3 \mu\text{A}$). This behavior is consistent with the corresponding increase observed for the charge transfer resistance by EIS measurement.

Supplementary Figure 9. Transient photocurrent response of MXene-derived $\gamma\text{-Fe}_2\text{O}_3/\text{Pt}/\text{TiO}_2$ microrobots (thermal annealing condition: 0 min at 550°C) before and after capturing nanoplastics (capture experiment conditions: 0.75 mg ml^{-1} microrobots, $\sim 10^{12}$ nanoplastics ml^{-1} , water at pH 3, UV-light irradiation for 5 min, microrobots collection using a neodymium magnet). The photocurrent was measured at 0 V vs. OCP by turning on and off the UV-light irradiation at time intervals of 10 s.

Supplementary Fig. 14 has been added to the Supplementary information and mentioned on page 21 of the manuscript as follows. The methods section has been updated accordingly.

“This observation is consistent with transient photocurrent response measurements, showing a microrobots’ photocurrent decrease after capturing nanoplastics (Supplementary Fig. 14).”

Response to the Reviewer #3

The authors report on microrobots of Fe₃O₄/Pt/TiO₂ for trapping and detecting nanoplastics from water. Thanks to the semiconductor TiO₂ and Pt coating, the resulted microrobots showed 6 degrees of freedom under UV light. The microrobots are positively charged and the nanoparticle of polymers are negatively charged and therefore they are attracted to each other, then the Fe₃O₄ can be utilized to collect the robots by magnetism for further detection studies. The authors used a number of characterization techniques (e.g., XRD, SEM, TEM, XPS, NTA, EIS) to study the synthesized microrobots and their behavior. The study is interesting, but it suffers from many weaknesses such as:

1- The use of MXene in this study is not justified at all considering the target is TiO₂ that is then get coated by Pt. It is a very expensive route to make TiO₂, while layered titanate can be made from much cheaper precursors than MXenes. It is not clear what is the value of using MXene in this study at all. A control sample using layered titanate should be used.

Author reply: We agree with the Reviewer's comment. The idea of this work is to develop a self-propelled light-powered microrobot to "on-the-fly" trap nanoplastics, allowing their preconcentration and further electrochemical detection. Indeed, it has been extensively demonstrated that active matter provides superior performance in the adsorption/degradation of water pollutants than static matter. The motion under light irradiation generally requires photocatalytic semiconducting microparticles, usually microspheres, which are asymmetrically coated by a metal layer to obtain the asymmetry necessary for the self-propulsion ability. As a consequence, half of the microspheres' surface is lost. Therefore, we opted for the exfoliated MXene, expecting that the sputtering process should not result in the loss of half of the microparticles' surface due to their multi-layered structure. To simplify the fabrication procedure and improve the results' reproducibility, we used a commercial exfoliated Ti₃C₂T_x MXene powder, which was *in situ* converted to TiO₂ to achieve a strong self-propulsion. However, when we designed this project, we did not know about the layered titanate. Hence, we wish to thank the Reviewer for this comment sincerely. In fact, layered titanate represents an ideal platform to design light-driven microrobots for environmental remediation. Stimulated by this comment, we tried to compare the performance of our MXene-derived microrobots with layered titanate. Since it is not commercially available, we tried to reproduce the protocol reported in published research articles. For example, we found a paper reporting the synthesis of layered titanate with a structure similar to our MXene-derived TiO₂ microparticles (Fig. R2), which present several slits accessible to the nanoplastics. Unfortunately, we could not reproduce the same layered titanate structure after several attempts. Additional unsuccessful tentatives have been conducted following other synthetic procedures. Since such comparison would require much more effort to reproduce the layered titanate structure, we prefer to better explain in the manuscript our choice to use the MXene as the primary building block of our microrobots. Also, we mentioned that layered titanate might represent an advantageous alternative due to its simpler preparation in the revised manuscript. For these reasons, we plan to use this fascinating material in our future projects. The following paragraph has been added on page 6 of the manuscript.

"The preservation of the multi-layered structure is crucial since it avoids sacrificing half of the microparticle's surface upon Pt deposition, as it occurs for smooth spherical particles. It should be noted that layered titanate presents a similar structure to the MXene-derived TiO₂ microparticles, also having the advantage of simpler and cheaper preparation, in principle.⁴⁵ Nevertheless, the utilization of commercial exfoliated Ti₃C₂T_x MXene and its oxidation into TiO₂ via thermal annealing was considered a more reproducible approach in this work."

Figure R2. SEM image of layered titanate reproduced from ref. [45].

Reference [43] has been included in the references list.

[45] Uekawa, N., Ono, Y. & Kojima, T. Synthesis of gluconate modified layered titanate particles using hydrolysis reaction of Ti alkoxide and characterization of their swelling behavior and structural color. *J. Sol-Gel Sci. Technol.* **85**, 48–58 (2018).

2- The use “MXene-based” should be removed from everywhere in the manuscript since based on XRD there is no MXene in these microrobots and they are mainly TiO₂. If the authors would like to keep MXene somewhere in the title, it should be “MXene-Derived Oxide”.

Author reply: We agree with the Reviewer’s comment. The term “MXene-based” has been changed to “MXene-derived oxide” or “MXene-derived TiO₂” throughout the manuscript, including the title, Supplementary information, and response letter. The new title is reported below.

“Trapping and detecting nanoplastics by MXene-derived oxide microrobots”

3- The justification for using multilayer MXene is not clear, the whole mechanism depends on surface charge, and multilayer MXene has a very low specific surface area and therefore the resulted oxide would have a low specific surface area too. The specific surface area of MXene and the derived microrobot should be measured

Author reply: As requested by the Reviewer, the specific surface area of the Ti₃C₂T_x MXene microparticles and MXene-derived γ -Fe₂O₃/Pt/TiO₂ microrobots was measured, finding 3.9 and 6.8 m² g⁻¹, respectively. The specific surface area of the MXene is in agreement with previous works, reporting values ranging from 0.5 to 9 m² g⁻¹ (references [43,44] in the manuscript). It is worth noting that the specific surface area value is strictly related to the MAX phase’s exfoliation process. Despite not being optimal, we preferred to use a commercially available exfoliated MXene for the sake of reproducibility. Instead, the larger surface area found for the MXene-derived microrobots is attributed to the thermal annealing process, which increased the roughness of the MXene’s surface due to the formation of TiO₂ nanoparticles. Indeed, an earlier study showed that the specific surface area of Ti₃C₂ MXene increases by ~4 times after a similar thermal treatment (reference [43] in the manuscript). In addition, Pt layer deposition and γ -Fe₂O₃ nanoparticles’ loading contributed to the surface area increase.

The following paragraph has been added on page 6 of the manuscript. The methods section has been updated accordingly.

“The specific surface area of the $Ti_3C_2T_x$ MXene and MXene-derived $\gamma-Fe_2O_3/Pt/TiO_2$ microrobots was measured, finding 3.9 and 6.8 $m^2 g^{-1}$, respectively. The former agrees with previous works, reporting values between 0.5 and 9 $m^2 g^{-1}$ for multi-layer Ti_3C_2 MXene.^{43,44} The larger surface area found for the microrobots is attributed to the thermal annealing process, converting the smooth $Ti_3C_2T_x$ surface into TiO_2 nanoparticles,⁴³ Pt layer deposition, and $\gamma-Fe_2O_3$ nanoparticles’ loading.”

Reference [44] has been included in the reference list.

[43] Shahzad, A. *et al.* Two-Dimensional $Ti_3C_2T_x$ MXene Nanosheets for Efficient Copper Removal from Water. *ACS Sustain. Chem. Eng.* **5**, 11481–11488 (2017).

The intercalation and delamination of the exfoliated $Ti_3C_2T_x$ MXene would significantly increase its surface area (reference [44] in the manuscript). However, half of the surface of each 2D MXene sheet would be sacrificed due to the Pt layer deposition required by the light-powered self-propulsion ability. Additionally, as discussed in the introduction section, this approach has already been explored, resulting in microrobots with a remarkably lower propulsive force (reference [41] in the manuscript).

We would like to point out that we agree with the Reviewer’s comment. In fact, the measured specific surface area values are relatively low. However, this characteristic does not alter the conclusions drawn in this work. Actually, the possibility of transferring the light-powered self-propulsion in 3D space, photocatalytic, magnetic, and pH-programmable surface charge properties of our MXene-derived microrobots into larger surface area materials promises to further enhance the nanoplastics trapping and detection concept here introduced. In this regard, the following paragraph has been added to the conclusions section on page 23 of the manuscript.

“Transferring the self-propulsion, photocatalytic, magnetic, and pH-programmable surface charge properties of the microrobots to larger surface area materials would significantly enhance the nanoplastics capture efficiency and, in principle, reduce the preconcentration time and improve the electrochemical sensor’s sensitivity.”

4- *The claim of nanoparticles trapped between layers should be revised considering. The nanoparticles can only access the external surface of the oxidized MXene and the macropores slits between the formerly multilayer stacks of MXenes. The volume of macropores is significantly reduced after oxidation as shown in figure S1b compared to S1a most. Also, the use of “layers” here is confusing since it relates to the 2D layers of MXenes where the interlayer spacing is <2 nm and can’t be accessed by the 50 nm plastic particles.*

Author reply: We thank the Reviewer for raising this point. We agree that the expression “nanoplastics trapped between layers” may be confusing. Therefore, this claim has been removed in the whole manuscript, specifying that nanoplastics are, indeed, trapped “on the microrobots’ surface, including the slits between the formerly MXene’s multi-layer stacks.” Moreover, we replaced the term “layered structure” with “multi-layered structure” throughout the manuscript for clarity. We did not modify the schemes since the nanoplastics’ capture is illustrated only in the microrobots’ surface and in the slits between multi-layer stacks.

5- *The lack of performance/behavior comparisons with other microrobots prepared differently make it very difficult for the reader to appreciate the advancement of this work. More comparisons with the literature is necessary for the discussion. Is it possible to re-use the microrobots?*

Author reply: The Reviewer is correct. On the other hand, such a comparison can not be provided since our work represents the first study on nanoplastics' removal using self-propelled nano/microrobots. Indeed, previous works (references [25,31-35] in the manuscript) were focused on polymers/micropastics' collection/degradation using microrobots. The removal efficiency has been generally measured by counting the number of micropastics before and after the treatment from optical microscopy images (references [32,33] in the manuscript). However, we believe that, due to the different sizes and concentrations of micropastics and nanoplastics, performance comparison with those works may be worthless.

Nonetheless, in reply to the following point raised by the Reviewer, the nanoplastics removal capacity q_t [mg g^{-1}] of our microrobots was calculated, allowing the performance comparison with "conventional" materials.

Moreover, as requested by the Reviewer, the reusability of the microrobots has been evaluated. Specifically, an aqueous solution at pH 3 with $\sim 6 \times 10^9$ nanoplastics ml^{-1} was treated with the microrobots for 5 min under UV-light irradiation. After the treatment, the nanoplastics' concentration in the solution was measured by NTA. Then, to induce the release of the captured nanoplastics, the microrobots were kept under vigorous agitation in an aqueous solution at pH 11 for 10 min. Finally, they were magnetically collected using a neodymium magnet and reused three times (four runs in total) following the same path. Supplementary Fig. 12, reporting the residual nanoplastics' concentration in the treated suspensions' after the four runs, indicates that microrobots' capture ability slightly decreases after each run. Therefore, the microrobots are reusable at least three times.

Supplementary Figure 10. Reusability test using the MXene-derived $\gamma\text{-Fe}_2\text{O}_3/\text{Pt}/\text{TiO}_2$ microrobots (thermal annealing condition: 0 min at 550°C). Capture experiments conditions: 0.75 mg ml^{-1} microrobots, 6×10^9 nanoplastics ml^{-1} , water at pH 3, UV-light irradiation for 5 min, microrobots collection using a neodymium magnet. After each run, the captured nanoplastics were released under vigorous agitation in water at pH 11 for 10 min. Error bars represent the standard deviation, $n = 3$ independent replicates.

Supplementary Fig. 12 has been added to the Supplementary information and commented on page 18 of the manuscript as follows. The methods section has been updated accordingly.

“The microrobots' reusability has been tested by several nanoplastics' capture and release cycles (Supplementary Fig. 12). It was found that they could be reused many times as their capture ability decreased slightly after each run.”

6- Any claim about removal capability should be supported by capacity of removal mg/g , the 97%

efficiency is meaningless without such quantification. Considering that surface area of multilayer MXene is only in the range of 10 m²/g, it will be surprising if the resulted composite has high removal capacity since it depends on surface charge. If the removal capacity can't be given, then such claims should be removed and the manuscript should focus on detection.

Author reply: As requested by the Reviewer, the nanoplastics' removal capacity q_t [mg g⁻¹] of the microrobots was calculated. Details of the calculation have been added to the methods section on page 27 of the manuscript as follows.

“The removal capacity q_t [mg g⁻¹] was calculated according to the following equation

$$q_t = \frac{C_0 - C_t}{m} V \quad (1)$$

where C_0 and C_t [mg ml⁻¹] are the nanoplastics concentration at 0 min and time t , respectively, V [ml] is the volume of the solution (1 ml), and m [g] is the sample's mass (7.5×10^{-4} g). Nanoplastics' concentration values measured by NTA [ml⁻¹] were converted to nanoplastics concentration values C_0 and C_t [mg ml⁻¹] using the estimated mass for the single nanoplastic (Supplementary Discussion 1).”

Supplementary Fig. 11 reports the calculated q_t values as a function of time. MXene-derived γ -Fe₂O₃/Pt/TiO₂ microrobots reached a removal capacity of 0.5 ± 0.1 mg g⁻¹ within 1 min treatment. We compared the microrobots' performance with other materials tested under similar conditions (e.g., contact time). Granular activated carbon showed a q_t value of ~0.3-0.7 mg g⁻¹ after 15 min under shaking [59]. Cellulose fibers got a q_t of 0.8-0.86 mg g⁻¹ after 5-120 min treatment under agitation [60]. Untreated coffee grains obtained a q_t of >2 mg g⁻¹ after 5 min under shaking [61]. Despite these materials having a larger surface area and being tested under external agitation/shaking, our microrobots almost exhibited a similar removal capacity in a shorter time due to the electrostatic capture mechanism and self-propulsion ability.

Supplementary Figure 11. Removal capacity q_t , calculated through Equation (1) in the manuscript, as a function of the treatment's duration (0, 1, 3, and 5 min) using γ -Fe₂O₃/Ti₃C₂T_x MXene microparticles under UV-light irradiation, MXene-derived γ -Fe₂O₃/Pt/TiO₂ microrobots (thermal annealing condition: 0 min at 550°C) with and without UV-light irradiation (other capture experiments conditions: 0.75 mg ml⁻¹ samples, 6×10^9 nanoplastics ml⁻¹, water at pH 3, samples' collection using a neodymium magnet). Error bars represent the standard deviation, $n = 3$ independent replicates.

Nevertheless, any claim of “high removal capacity” has been removed from the manuscripts. Supplementary Fig. 11 has been included in the Supplementary information and commented on page 18 of the manuscript as follows.

“The microrobots’ removal capacity q_t [mg g^{-1}] was calculated and plotted in Supplementary Fig. 11 to allow a straightforward comparison with “conventional” materials tested under similar conditions in recent studies. The microrobots reached a q_t of $0.5 \pm 0.1 \text{ mg g}^{-1}$ within 1 min treatment. This value is close to that reported for other materials, such as granular activated carbon ($\sim 0.3\text{-}0.7 \text{ mg g}^{-1}$ after 15 min),⁵⁹ cellulose fibers ($0.8\text{-}0.86 \text{ mg g}^{-1}$ after 5-120 min),⁶⁰ and untreated coffee grains ($>2 \text{ mg g}^{-1}$ after 5 min),⁶¹ despite having a larger surface area and being utilized under external agitation.”

References [59-61] have been comprised in the reference list.

[59] Ramirez Arenas, L., Ramseier Gentile, S., Zimmermann, S. & Stoll, S. Nanoplastics adsorption and removal efficiency by granular activated carbon used in drinking water treatment process. *Sci. Total Environ.* **791**, 148175 (2021).

[60] Batool, A. & Valiyaveetil, S. Surface functionalized cellulose fibers – A renewable adsorbent for removal of plastic nanoparticles from water. *J. Hazard. Mater.* **413**, 125301 (2021).

[61] Yen, P. L., Hsu, C. H., Huang, M. L. & Liao, V. H. C. Removal of nano-sized polystyrene plastic from aqueous solutions using untreated coffee grounds. *Chemosphere* **286**, 131863 (2022).

7- The EIS data in fig 6a is not presented correctly. Nyquist plot shape is meaningless unless the graph is square with equivalent axes.

Author reply: We agree with the Reviewer’s comment. As requested, the Nyquist plot in Fig. 6(a) in the manuscript has been changed into a squared graph with equivalent axes, as reported below.

Figure 12. Nanoplastics’ detection by electrochemical impedance spectroscopy (EIS) technique after the preconcentration with the MXene-derived $\gamma\text{-Fe}_2\text{O}_3/\text{Pt}/\text{TiO}_2$ microrobots. (a) Nyquist plots showing the impedance real ($\text{Re}(Z)$) and imaginary ($-\text{Im}(Z)$) parts as a function of the frequency for a bare screen-printed electrode (SPE), an SPE exposed to $\sim 10^{12}$ nanoparticles ml^{-1} suspension, an SPE after MXene-derived $\gamma\text{-Fe}_2\text{O}_3/\text{Pt}/\text{TiO}_2$ microrobots’ (thermal annealing condition: 0 min at 550°C) loading, and an SPE after loading microrobots with captured nanoparticles (capture experiment conditions: 0.75 mg ml^{-1} microrobots, $\sim 10^{12}$ nanoparticles ml^{-1} , water at pH 3, UV-light irradiation for 1 min, microrobots collection using a neodymium magnet). The EIS measurements were performed in a $10 \text{ mM Fe}(\text{CN})_6^{4-/3-}$ aqueous solution at 0 V vs. ref. with a 10 mV superimposed sinusoidal root-mean-square voltage in the frequency range $10^5\text{-}10^0 \text{ Hz}$. The lines represent fits to the data.

8- While one can assume that all of the results in the paper relate to the 0 min thermal annealing condition, since the authors note that it preserved the layered structure best, it is not clear that all subsequent figures/results relate to the 0 min annealing condition.

Author reply: The Reviewer is right. Therefore, the following sentence has been added at the end of the discussion about the thermal annealing process optimization on page 6 of the manuscript.

“For this reason, all results presented in the following relate to the 0 min annealing condition.”

Moreover, the following line has been included in all figures’ captions involving the “optimal” MXene-derived TiO₂ microparticles or MXene-derived γ -Fe₂O₃/Pt/TiO₂ microrobots.

“(thermal annealing condition: 0 min at 550°C)”

Furthermore, for the sake of clarity, in the captions of figures involving nanoplastics’ capture experiments, we added a line summarizing the capture experiments conditions, for example:

“(capture experiment conditions: 0.75 mg ml⁻¹ microrobots, ~10¹² nanoplastics ml⁻¹, water at pH 3, UV-light irradiation for 1 min, microrobots collection using a neodymium magnet)”

Remarks to the Editor and Reviewers

During the revision process, we noticed small mistakes in the XPS spectra of $\text{Ti}_3\text{C}_2\text{T}_x$ MXene in Fig. 1(e) and in the SEM images in Fig. 5(c) in the manuscript, which have been corrected in the revised version of our manuscript. It is worth noting that such corrections do not influence the discussion in the manuscript. Supplementary Table 1, reporting the XPS peak fitting results, has been updated accordingly.

REVIEWERS' COMMENTS

Reviewer #1 (Remarks to the Author):

The authors have explained in detail the role of Pt coating on MXene-derived oxide microrobots. The corresponding experiments were supplemented to confirm the reliability of the capture and detection of nanoplastics based on the γ -Fe₂O₃/Pt/TiO₂ microrobots. In addition, the author supplemented the 2D and 3D motion mechanism of γ -Fe₂O₃/Pt/TiO₂ microrobots under UV-light irradiation. Therefore, I recommend this article for publication in Nature Communications.

Reviewer #3 (Remarks to the Author):

The failure to compare the TiO₂ from using MXene with other oxide is a significant weakness. Layered titanites are well explored in the literature and should be produced very easily.

The authors addressed my other comments.

Responses to the comments of the Reviewers

Response to the Reviewer #1

The authors have explained in detail the role of Pt coating on MXene-derived oxide microrobots. The corresponding experiments were supplemented to confirm the reliability of the capture and detection of nanoplastics based on the γ -Fe₂O₃/Pt/TiO₂ microrobots. In addition, the author supplemented the 2D and 3D motion mechanism of γ -Fe₂O₃/Pt/TiO₂ microrobots under UV-light irradiation. Therefore, I recommend this article for publication in Nature Communications.

Author reply: We thank the Reviewer for this comment and for recommending the article for publication in Nature Communications.

Response to the Reviewer #3

The failure to compare the TiO₂ from using MXene with other oxide is a significant weakness. Layered titanites are well explored in the literature and should be produced very easily. The authors addressed my other comments.

Author reply: We thank the Reviewer for this comment and for recommending the article for publication in Nature Communications.